# Learning Sub-Second Routing Optimization in Computer Networks requires Packet-Level Dynamics

**Andreas Boltres** *andreas.boltres@partner.kit.edu*
*Autonomous Learning Robots,*
*Karlsruhe Institute of Technology*
*SAP SE*

**Niklas Freymuth** *niklas.freymuth@kit.edu*
*Autonomous Learning Robots,*
*Karlsruhe Institute of Technology*

**Patrick Jahnke** *pj@turba.ai*
*Turba AI*

**Holger Karl** *holger.karl@hpi.de*
*Internet-Technology and Softwarization,*
*Hasso-Plattner-Institut Potsdam*

**Gerhard Neumann** *gerhard.neumann@kit.edu*
*Autonomous Learning Robots,*
*Karlsruhe Institute of Technology*

**Reviewed on OpenReview:** *https://openreview.net/forum?id=H95g8UpYKY*

## Abstract

Finding efficient routes for data packets is an essential task in computer networking. The optimal routes depend greatly on the current network topology, state and traffic demand, and they can change within milliseconds. Reinforcement Learning can help to learn network representations that provide routing decisions for possibly novel situations. So far, this has commonly been done using fluid network models. We investigate their suitability for millisecond-scale adaptations with a range of traffic mixes and find that packet-level network models are necessary to capture true dynamics, in particular in the presence of TCP traffic. To this end, we present *PackeRL*, the first packet-level Reinforcement Learning environment for routing in generic network topologies. Our experiments confirm that learning-based strategies that have been trained in fluid environments do not generalize well to this more realistic, but more challenging setup. Hence, we also introduce two new algorithms for learning sub-second Routing Optimization. We present *M-Slim*, a dynamic shortest-path algorithm that excels at high traffic volumes but is computationally hard to scale to large network topologies, and *FieldLines*, a novel next-hop policy design that re-optimizes routing for any network topology within milliseconds without requiring any re-training. Both algorithms outperform current learning-based approaches as well as commonly used static baseline protocols in scenarios with high-traffic volumes. All findings are backed by extensive experiments in realistic network conditions in our fast and versatile training and evaluation framework.[1]

## 1 Introduction

Routing data packets efficiently is an essential task in computer networks. A well-working routing mechanism maximizes service quality and minimizes operational cost. Several conditions make network traffic routing a

---

[1]Code available via project webpage: `https://alrhub.github.io/packerl-website/`

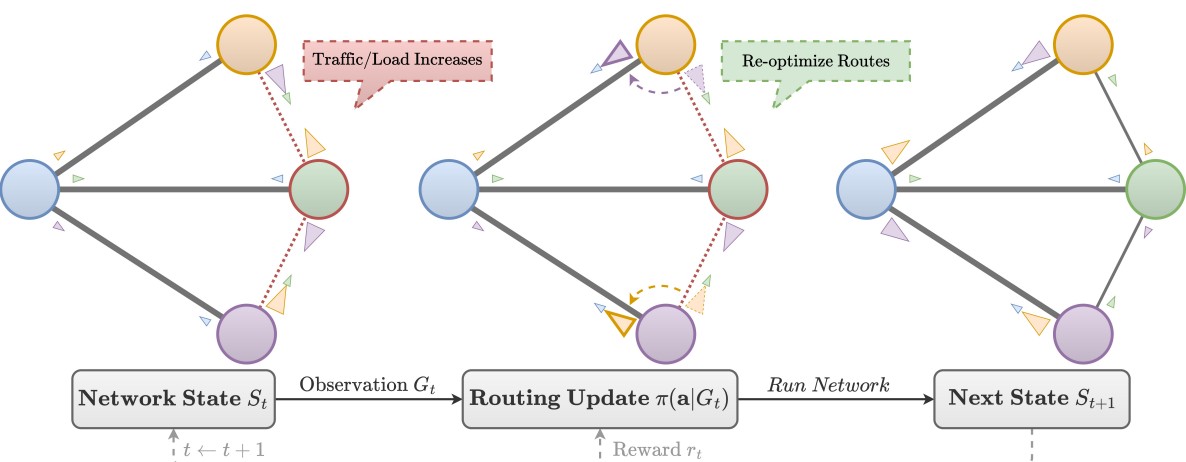

Figure 1: Re-optimizing packet routes based on the network topology and current utilization and load values can minimize congestion, delay and packet drops: Here, the longer but higher-capacity path (thicker edges) is preferred to the shorter path when traffic spikes for the orange (top) and purple (bottom) node, causing the algorithm to re-route traffic over the blue (left) node instead of the green (right) one.

complex problem: i) It can be optimized with respect to various performance metrics, like packet delay or loss of data (Wang et al., 2008). ii) Traffic demands are often highly volatile and unpredictable, e.g. in datacenter or content delivery networks (Alizadeh et al., 2014; Wendell & Freedman, 2011). iii) Network topologies are often subject to unexpected changes like link and switch failures (Gill et al., 2011; Markopoulou et al., 2008; Turner et al., 2010). iv) The space of possible routing decisions grows exponentially with network size. Traffic Engineering (TE) is a highly active research area that tackles this problem (Farrel, 2024) by means of regular monitoring and control. Within TE, a core problem is Routing Optimization (RO). In various network setups, TE algorithms claim to produce optimal or near-optimal routing at sensible cost (Mendiola et al., 2016). Yet finding optimal routes given a network configuration and traffic information is often computationally intractable (Xu et al., 2011). Also, to deal with the uncertainty about future traffic conditions, conventional TE methods are limited to optimize for previously observed or speculated future traffic. They may perform poorly even when the observed traffic is only slightly off (Valadarsky et al., 2017). As a remedy, researchers have turned to data-driven optimization via deep Reinforcement Learning (RL) (Xiao et al., 2021).

With deep RL, routing policies can use learned representations of network states to provide routing decisions for possibly novel situations. These states are conditioned on the encountered *network scenario*, i.e., the network topology and configuration and their changes over time, as well as traffic demands. Contributors of RL for TE have presented performance improvements in widely varying experiment settings. They range from a handful of experiments on real or emulated testbed networks (Guo et al., 2022; Huang et al., 2022; Pinyoanuntapong et al., 2019) to small ranges of network scenarios evaluated in simulations with more abstract network models (Bernárdez et al., 2023; Xu et al., 2023). Covering the variety of network scenarios that is required to thoroughly evaluate RL-based RO approaches (Zhang et al., 2018) can become prohibitively expensive when using real testbed networks (Sherwood et al., 2010). Consequently, building realistic yet versatile simulation environments is particularly important.

Out of the publicly available environments, only the one by Bernárdez et al. (2023) supports arbitrary network topologies and traffic patterns. But it uses a fluid-based network model, i.e., a model that treats traffic as flows in a flow network. Such models abstract away from packet-level interactions as encountered in the real world, e.g. in TCP traffic, where sending and forwarding dynamics are very different from flow distributions in flow networks. Hence, it is unclear how models trained in such environments perform in more realistic scenarios, where the traffic patterns are shaped by the dynamics of transport protocols like TCP. Besides the increased realism, including packet-level interactions in the environment also permits routing control on a finer temporal scale. Related work has indeed recognized the importance of sub-second RO in an increasing number of network scenarios (Gay et al., 2017a), stressing the need for such packet-level environments. To address these shortcomings of fluid-based simulation, we hence introduce a new packet-level training and

evaluation environment *PackeRL* that is tailored to learning sub-second RO. It realistically simulates network scenarios with versatile network topologies, traffic data, and various transport protocols.

Current RL-powered RO approaches either need to compute shortest paths multiple times for every re-optimization step (Bernárdez et al., 2023), which does not scale to sub-second RO for large network architectures, or select next-hop neighbors separately at each routing node (Valadarsky et al., 2017; Pinyoanuntapong et al., 2019; You et al., 2022; Guo et al., 2022; Bhavanasi et al., 2022), which does not generalize to arbitrary network topologies without the need for re-training or adjusting the policy architecture. This paper presents two new RL-powered approaches for sub-second RO: *M-Slim* and *FieldLines*. *M-Slim* is a scalable RO policy that only requires, per update, a single All Pairs Shortest Paths (APSP) calculation for obtaining a new routing strategy, unlike existing methods that require many optimizations (Bernárdez et al., 2023). While this reduces the time required to re-optimize routing significantly, the computation time still grows cubically with the number of network nodes when using the Floyd-Warshall algorithm (Cormen et al., 2022). *FieldLines* is a novel next-hop selection RO approach that leverages the permutation equivariance properties of its Graph Neural Network (GNN) architecture (Bronstein et al., 2021) to generalize to unseen network topologies. It is the first next-hop selection design that provides competitive RO performance in packet-level environments on any kind and scale of network topology, while only requiring training on small networks with up to 10 nodes.

In summary, our contributions are as follows: i) *PackeRL* is the first packet-level simulation environment RL-powered RO that supports arbitrary yet realistic network topologies and traffic data, including UDP and TCP traffic. ii) Using *PackeRL*, we show that an existing RL-based RO policy trained in a fluid-based environment like MAGNNETO (Bernárdez et al., 2023) cannot cope with packet-level network dynamics and performs significantly worse than common static shortest-paths routing strategies. This motivates the need for packet-level training environments like *PackeRL*. iii) We present *M-Slim*, a novel RL-based shortest path RO method that scales to sub-second RO and is trainable in our packet-level environment *PackeRL*. It outperforms static shortest-path routing strategies by a significant margin. iv) We present *FieldLines*, a novel next-hop routing policy that provides competitive performance for traffic-intense scenarios and does not suffer from the computational limitations of shortest-path based policies.

## 2  Related Work

Researchers working on conventional TE approaches have recognized the need for standardized evaluation environments: REPETITA (Gay et al., 2017b) aims at fostering reproducibility in TE optimization research, and the goal of YATES (Kumar et al., 2018) is to facilitate rapid prototyping of TE systems. While they are not deliberately designed with RL in mind, it is possible to extend them to support training and evaluating RL-based routing policies. These frameworks are limited by their abstract network model which does not model packet-level interactions. Our experiments demonstrate that RL routing policies trained in such environments perform significantly worse when tasked to route in more realistic packet-level environments.

The simulation frameworks RL4Net (Xiao et al., 2022), its successor RL4NET++ (Chen et al., 2023) and PRISMA (Alliche et al., 2022) are perhaps the closest relatives to *PackeRL*. They provide the same kind of closed interaction loop between RL models and algorithms in Python and packet-level network simulation in *ns-3*. However, in contrast to *PackeRL*, out-of-the-box support for arbitrary network topologies including link datarates and delays is left to the user, and simulation is limited to constant bit-rate traffic between all pairs of nodes. The latter does not apply to PRISMA, which however only simulates UDP traffic. Moreover, Figure 7 shows that *PackeRL* runs simulation steps several times faster than the numbers reported in Chen et al. (2023). This may be due to *PackeRL* leveraging the shared-memory interface of *ns3-ai* for communication between learning and simulation components, instead of inter-process communication via ZeroMQ (Hintjens, 2013). As the authors of *ns3-ai* noted in Yin et al. (2020), this drastically cuts communication times between learning and simulation components.

A common approach to learn RO with deep RL is to infer link weights that are used to compute routing paths (Stampa et al., 2017; Pham et al., 2019; Sun et al., 2021; Bernárdez et al., 2023). Out of the existing approaches, only MAGNNETO (Bernárdez et al., 2023) can generalize to unseen topologies by using GNNs in their policy designs. The caveat of MAGNNETO is the iterative process of $\Theta(E)$ steps required to optimize

routing for a single Traffic Matrix (TM)[2]. Each iteration step, its actions denote a set of links whose weight shall be incremented for the upcoming optimization iteration. These actions don't lend to paths directly. Instead, the link weights for path computation are obtained only after finishing the optimization process of $\Theta(E)$ steps. Each step requires an APSP computation with a computational complexity of $O(V^3)$ when using Floyd-Warshall or $O(V^2 \log V + EV)$ when using Johnson-Dijkstra (Cormen et al., 2022). Consequently, MAGNNETO is not able to provide sub-second RO in most networks, as the results in Figure 7 show.

Instead of computing shortest paths for RO, some deep RL approaches are trained to select next-hop neighbors directly (Valadarsky et al., 2017; Pinyoanuntapong et al., 2019; You et al., 2022; Guo et al., 2022; Bhavanasi et al., 2022). However, these policies are not designed to generalize to arbitrary topologies and to handle topology changes, either because their GNN architectures require a (re-)training process for each topology (Bhavanasi et al., 2022; Mai et al., 2021; You et al., 2022), or because their non-GNN architecture ties them to specific topologies (Valadarsky et al., 2017; Pinyoanuntapong et al., 2019; Guo et al., 2022).

A small portion of the existing RL-powered RO approaches has open-sourced their training and evaluation environments (Stampa et al., 2017; Bernárdez et al., 2023; Xu et al., 2023). These environments either employ the same limiting model abstractions as REPETITA and YATES (Bernárdez et al., 2023; Xu et al., 2023), or, if they leverage network simulator backends, they only support a limited subset of network scenarios (Stampa et al., 2017). Therefore, they are not suited to become reference frameworks for training and evaluation.

Finally, recent related work has proposed learned network models as replacements for packet-based network simulation (Zhang et al., 2021; Yang et al., 2022; Wang et al., 2022; Ferriol-Galmés et al., 2023). In general, such models receive network topologies, traffic and routing as input, and consult a trained model to predict performance metrics like packet delay and link utilization. While these models promise smilar accuracy at lower computational cost, the reported gaps in accuracy (Ferriol-Galmés et al., 2023) suggest that these frameworks cannot fully replace the packet-level simulation offered by *PackeRL*.

In summary, there exists no framework for RL-powered RO approaches that offers both a realistic simulation backend, and a comprehensive toolset for training and evaluation on wide ranges of realistic network scenarios. Besides, there is no experimental evidence for why such packet-level frameworks are even needed, for instance by exposing the limits of routing policies trained in more abstract environments. Finally, none of the existing next-hop selection RO approaches work on arbitrary and changing topologies without re-training or architectural adjustments, and the shortest-path based algorithms that do generalize across topologies are computationally too expensive to provide sub-second routing. We believe that *PackeRL*, our experimental results obtained for our shortest-path routing policy *M-Slim* in *PackeRL*, and our new next-hop policy design *FieldLines* close these gaps.

## 3 Preliminaries and Problem Formulation

We consider wired Internet Protocol (IP) networks using the connection-less User Datagram Protocol (UDP) (Postel, 1980) and the connection-based Transmission Control Protocol (TCP) (Eddy, 2022). TCP in particular is responsible for the majority of today's internet traffic (Schumann et al., 2022).

Routing selects paths in a network along which data packets are forwarded. To determine the best paths, the Routing Protocol (RP)'s algorithm uses the network topology as input. For example, Open Shortest-Path First (OSPF) (Moy, 1997) propagates each link's data rate such that every router knows the entire network graph, annotated by data rate. Then, every router uses Dijkstra to compute shortest paths, where by default the path cost is the sum of the inverse of link data rates (Section B.5 explains the path cost calculation process). Several network performance indicators exist, such as **goodput** (i.e. the bitrate of traffic received at the destination), **latency/delay** (the time it takes for data to travel from the source to the destination), **packet loss** (the percentage of data that is lost during transmission), or **packet jitter** (the variability in packet arrival times, which can affect the quality of real-time applications or lead to out-of-order data arrival). Achieving a favorable trade-off between these performance metrics is non-trivial, not least because their importance may vary depending on the type/use case of network and the traffic characteristics.

---

[2]A TM contains average traffic flows between pairs of nodes $i, j \in V \times V$ in a square matrix representation.

### 3.1 Routing Optimization as an RL Problem

We formalize RO as a Markov Decision Process with the tuple $\langle \mathcal{S}, \mathcal{A}, \mathcal{T}, r \rangle$, splitting the continuous-time network operation into time slices of length $\tau_{\text{sim}}$. The space of *network states* $\mathcal{S}$ consists of attributed graphs with global, node- and edge-level performance and load values, as well as topology characteristics like link datarate and packet buffer size. We model network states as directed graphs $S_t = (V_t, E_t, \mathbb{X}_{V_t,t}, \mathbb{X}_{E_t,t}, \mathbf{x}_{u,t})$ with nodes $V_t$ and edges $E_t$ at step $t$. Node and edge features are given by $\mathbb{X}_{V_t,t} = \{\mathbf{x}_{v,t} \in \mathbb{R}^{d_{V_t}} \mid v \in V_t\}$ and $\mathbb{X}_{E_t,t} = \{\mathbf{x}_{e,t} \in \mathbb{R}^{d_{E_t}} \mid e \in E_t\}$ respectively, and $\mathbf{x}_{u,t} \in \mathbb{R}^{d_U}$ denotes global features that are shared between all nodes. Sections A.4 and B.4 contain details on how network states are obtained in our framework. The action $\mathbf{a}_t \in \mathcal{A}$ consists of a next-hop neighbor selection $v \in \mathcal{N}_u$ per routing node $u \in V_t$ for each possible destination node $z \in V_t$, or formally: $\mathbf{a}_t = \{(u, z) \mapsto v \mid u, v, z \in V_t, v \in \mathcal{N}_u\}$. These destination-based routing actions are valid for all packets processed in the upcoming timestep, i.e. the actions are taken in the control plane (Mestres et al., 2017). The transition function $T : \mathcal{S} \times \mathcal{A} \to \mathcal{S}$ evolves the current network state using the induced routing actions to obtain a new network state. Its transition probabilities are unknown because it depends on the upcoming traffic demands, which are often unpredictable in practice (Wendell & Freedman, 2011). Finally, $r : \mathcal{S} \times \mathcal{A} \to \mathbb{R}$ is a global reward function which assesses the fit of a routing action in the given network topology and state. Here, we use the global goodput, measured as MB received per step, as our reward function. Our goal is to find a policy $\pi : \mathcal{S} \times \mathcal{A} \to [0,1]$ that maximizes the return, i.e., the expected discounted cumulative future reward $J_t := \mathbb{E}_{\pi(\mathbf{a}|\mathbf{s})}\left[\sum_{k=0}^{\infty} \gamma^k r(\mathbf{s}_{t+k}, \mathbf{a}_{t+k})\right]$. Thus, in our default setting, an optimal policy maximizes the long-term global goodput. Despite its simplicity, this optimization objective provides competitive results. Section C.3 shows that optimizing for different objectives, including multi-component objectives, does not improve results consistently.

Next, we introduce our packet-level simulation framework *PackeRL* which builds upon the above formalism.

## 4 *PackeRL*: An Overview

*PackeRL* is a framework for training and evaluating deep RL approaches that route packets in IP computer networks. It interfaces the discrete-event network simulator *ns-3* (Henderson et al., 2008) for repeatable and highly configurable RO experiments with realistic network models. It provides a *Gymnasium*-like interface (Towers et al., 2023) to the learning algorithm and provides near-instantaneous communication between learning algorithm and simulation backend by using the shared memory extension *ns3-ai* (Yin et al., 2020). *PackeRL* advances RL research for RO in the following ways:

- It supports optimization for several common objectives that can be combined with adjustable weightings. Nonetheless, both RL and non-RL approaches alike can be evaluated with respect to a range of performance metrics. While we optimize our approach for goodput by default, we show results for alternative optimization objectives in Section C.3.

- It provides access to an extensive range of network scenarios. Section 4.1 provides further information.

- It implements a closed interaction loop between RL policy and network simulation, using In-Band Network Telemetry (Kim et al., 2015) to monitor the network state and provide state snapshots $S_t$. The time period simulated per environment step can be arbitrarily large or small, and thus, unlike most existing frameworks, it supports online RO experiments with or without RL.

- It can be used to train RL routing policies within a few hours and evaluate them within a few minutes. Thus we dispel the concerns raised by related work that packet-based environments are too slow (Ferriol-Galmés et al., 2023) or too complex to implement (Kumar et al., 2018).

We provide a detailed explanation of *PackeRL* in Section A. Section A.1 explains the structure of computer networks in *ns-3*, Section A.2 expands on the interaction loop between RL policy and environment, Section A.3 clarifies how the actions $\mathbf{a}_t$ are installed in the simulated network, and Section A.4 provides details on how network states $S_t$ are obtained from the network simulation. Finally, general simulation parameters are explained in Section B.1.

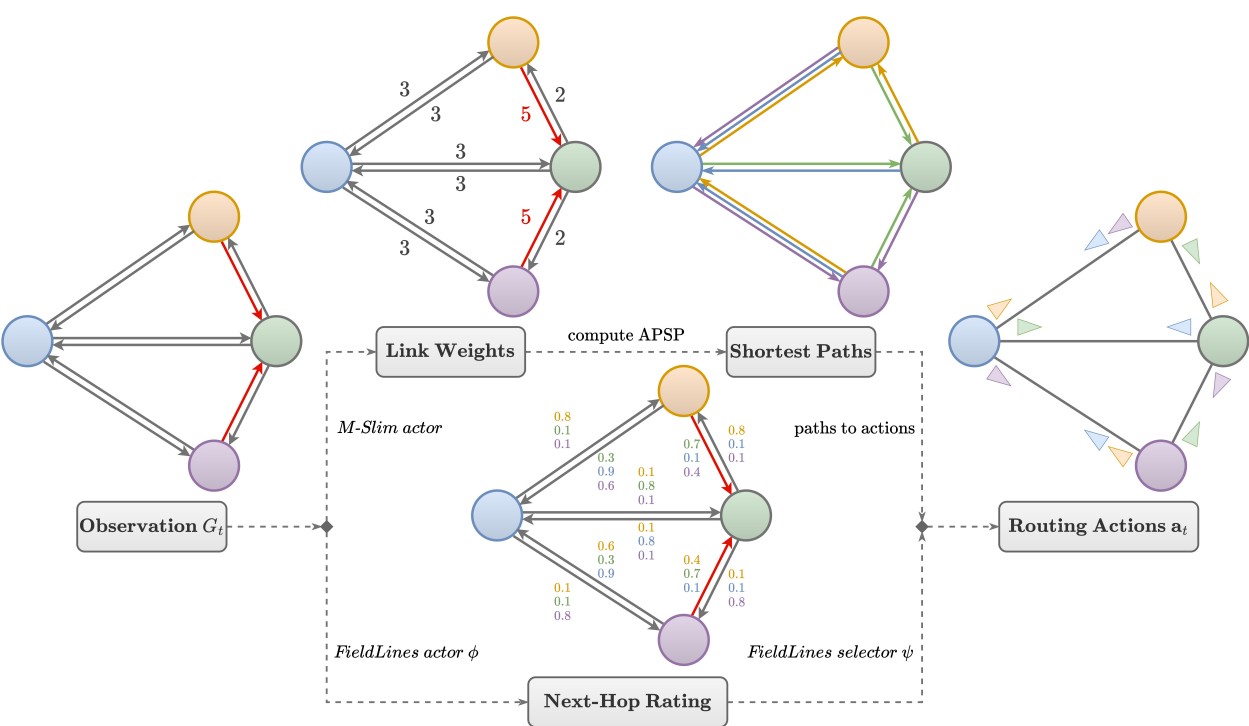

Figure 2: Example of how the learnable policies *M-Slim* and *FieldLines* obtain routing actions $\mathbf{a}_t \in \mathcal{A}$ from network states $S_t$. The red edges denote highly loaded data pathways, e.g. due to full packet buffers. The actor of *M-Slim* outputs link weights that are used to calculate routing paths. These routing paths are then broken down into individual next-hop neighbor selections per destination node $v \in V$ and routing node $u \in V$ to fit the definition of the action space $\mathcal{A}$. *FieldLines* uses its *actor* module $\phi$ to obtain next-hop ratings per edge and destination node, illustrated by the respective colors of the rating values. The *selector* module $\psi$ then uses these ratings to select next-hop neighbors per destination and routing node.

## 4.1 Network Scenario Generation with *synnet*

*PackeRL* offers versatile simulation conditions via *synnet*, a standalone module for *network scenario* generation. In *synnet*, network scenarios consist of the network topology and a set of events. The network topology consists of the graph of routing nodes, links between them, as well as parameters like link data rate or delay and packet buffer size. For generating the topologies, we use random graph models commonly found in the literature (Barabási, 2009; Erdős et al., 1960; Watts & Strogatz, 1998). Events can be of two types: Traffic demand events contain an arrival time, demand size and type (UDP vs. TCP). We use random models to generate demand arrival times and volumes, such that the generated network traffic resembles observed real-world traffic patterns (Benson et al., 2010). Link failure events consist of a failure time and the edge that is going to fail. Here, too, we use random models for link failure times that resemble the patterns found in operative networks (Bogle et al., 2019). Sections A.5 and A.6 provide more details on the scenario generation process as well as examples, while Sections B.2 and B.3 contain the parameters used for the random models.

## 5 Policy Designs for Routing in Packet-Level Environments

This section introduces two RL policy designs for RO trainable in *PackeRL*, namely our next-hop selection policy *FieldLines* and our adaptation of MAGNNETO which we call *M-Slim* (short for *MAGNNETO-Slim*). Our results in Section 7 show that both *M-Slim* and *FieldLines* clearly outperform MAGNNETO, underlining the benefit of learning to route in *PackeRL*. Nevertheless, they use different approaches to obtain routing actions $\mathbf{a}_t$ from the given network state $S_t$. As illustrated in Figure 2, *M-Slim* adopts MAGNNETO's approach of computing routing paths from inferred link weights, while *FieldLines* directly selects next-hop

| | EIGRP | FieldLines (ours) | M-Slim (ours) | MAGNNETO | OSPF | Random (LW) | Random (NH) | | EIGRP | FieldLines (ours) | M-Slim (ours) | MAGNNETO | OSPF | Random (LW) | Random (NH) |
|---|---|---|---|---|---|---|---|---|---|---|---|---|---|---|---|
| | | | TCP/UDP, medium traffic | | | | | | | | TCP, high traffic | | | | |
| Action Fluctuation | 0% | +3.64% (2.13, 5.8) | +8.96% (7.47, 10.19) | +8.75% (5.02, 13.41) | 0% | +22.25% | +61.65% | | 0% | +4.4% (2.7, 7.02) | +9.21% (8.57, 10.04) | +8.56% (7.32, 10.98) | 0% | +21.1% | +60.25% |
| Received (↑) | 35.38MB | +0.28% (35.38, 35.52) | -0.37% (34.71, 35.67) | -5.58% (31.54, 35.08) | -1.26% | -23.14% | -78.58% | | 61.7MB | +1.92% (62.79, 62.94) | +3.25% (62.83, 64.33) | -4.48% (53.07, 63.18) | -2.62% | -20.96% | -88.63% |
| Avg. Delay (↓) | 7.2ms | -2.24% (7.02, 7.05) | -3.37% (6.94, 6.98) | -1.66% (7.0, 7.19) | +2.57% | +5.38% | +232.54% | | 7.53ms | -1.84% (7.37, 7.41) | -2.87% (7.28, 7.33) | -0.82% (7.27, 7.78) | +1.83% | +2.46% | +170.26% |
| Dropped (↓) | 1.25% | +0.14% (1.37, 1.4) | +0.15% (1.31, 1.49) | +0.43% (1.36, 2.01) | -0.02% | +1.29% | +16.66% | | 5.74% | -0.14% (5.51, 5.68) | -0.23% (5.42, 5.66) | +0.44% (5.65, 6.89) | +0.16% | +1.93% | +20.76% |

Figure 3: Results on the *nx–XS* topology preset, displayed per approach and performance metric. Cells show the mean values over 100 evaluation episodes in the first line, and min and max values across random seeds in the second line. Values are relative to Enhanced Interior Gateway Routing Protocol (EIGRP). The stark contrast between random and learned routing shows that efficient routing is not a trivial task, and using RL to learn it is very beneficial. Both our approaches outperform the shortest-path baselines in high-traffic scenarios, and the difference in performance to MAGNNETO shows that learning to route in packet-based environments is important.

neighbors per destination node $v \in V$ and routing node $u \in V$. Section 5.2 explains how this circumvents the need for computing shortest paths on every re-optimization.

The information available in the RO problem can be represented as graphs with node, edge and global features. Thus, GNNs are highly suitable models because their permutation equivariance enables generalization to arbitrary network topologies. While MAGNNETO uses Message Passing Neural Networks (MPNNs) (Gilmer et al., 2017) for its actor module, we use a variant valled Message Passing Networks (MPNs) (Sanchez-Gonzalez et al., 2020; Freymuth et al., 2024) that supports node, edge, and global features. Using Multilayer Perceptrons (MLPs) $f^l$, initial features $\mathbf{x}_v^0$ and $\mathbf{x}_e^0$ with $e = (v, u) \in E$, the $l$-th step is given as

$$\mathbf{x}_e^{l+1} = f_E^l(\mathbf{x}_v^l, \mathbf{x}_e^l), \qquad \mathbf{x}_v^{l+1} = f_V^l(\mathbf{x}_v^l, \bigoplus_{e=(v,u)} \mathbf{x}_e^{l+1})$$

For the permutation-invariant aggregation $\oplus$, we use a concatenation of the features' mean and minimum. We provide implementation details on the MPNN architecture of MAGNNETO and the MPN architecture used by *M-Slim* and *FieldLines* in Appendix B.8.

## 5.1 *M-Slim*: Learning Link-Weight Optimization in Packet-Level Simulation

Section 2 states that MAGNNETO's iterative process for obtaining routing paths is too slow to warrant sub-second RO. In fact, the inference times reported in Figure 7 are too high even to follow our training protocol in *PackeRL* because it involves a larger number of temporally fine-grained interaction steps. Reducing inference time is required to enable training in *PackeRL*, and to this end we present our MAGNNETO adaptation called *M-Slim*. While the output of MAGNNETO's actor architecture specifies a set of links whose weights shall be incremented, in *M-Slim* we interpret this output as link weights directly. This reduces the amount of model inference steps and APSP computations per routing update from $\Theta(E)$ to 1 and enables sub-second routing re-optimization in larger network topologies.

The change in interpretation of the actor's output requires further design adjustments. To account for exploration during training rollouts, we treat the actor output as the mean $\mu_t$ for a diagonal Gaussian $\mathcal{N}(\mu_t, \sigma_{M\text{-}Slim})$ from which we then sample the actual link weights. $\sigma_{M\text{-}Slim}$ is a learnable parameter. During evaluation, the values of $\mu_t$ are used as link weights directly. Finally, we apply the Softplus function (Zheng et al., 2015) to the output of the actor module to ensure the values are positive and thus usable as link weights. Concerning the architecture of the actor module, we replace the MPNN design used by MAGNNETO by an MPN-based design with $L = 2$ steps and parameters as detailed in Appendix B.8 to reduce model complexity. As Figure 17 of Appendix C.2 shows, compared to when using MAGNNETO's MPNN architecture for *M-Slim*, the MPN design further decreases inference time without compromising on performance. Otherwise, *M-Slim* uses the same input and output representations as MAGNNETO: They receive the latest Link Utilization

(LU) values and past link weights as input and operate on the *Line Digraph* representation to obtain link weights as node features. Section B.6 further explains Line Digraphs.

### 5.2 *FieldLines*: Fast Next-Hop Routing in any Network Topology

*M-Slim* greatly reduces computation time per re-optimization step, but it still requires one APSP pass per routing update as illustrated in Figure 2. A single such computation overshadows the computational complexity term $O(VD)$ of GNN inference (Alkin et al., 2024) for a maximum node degree $D$ as the network grows in scale, making the shortest-paths computation the computational bottleneck. We thus turn to our next-hop selection RO approach *FieldLines* to further reduce inference times: It utilizes the results of one initial APSP computation and only requires re-computing APSP when the network topology changes.

*FieldLines* consists of an *actor* module $\phi$ and a *selection* module $\psi$: The *actor* module $\phi : \mathcal{S} \to \mathbb{R}^{|V| \times |E|}$ first creates an embedding of the network state that it then uses to provide numerical values $\phi_{z,e}$ for each possible combination of network edge $e \in E$ and destination node $z \in V$. We implement $\phi$ as an MPN with $L = 2$ steps and parameters as described in Appendix B.8. Intuitively, the output values $\phi_{z,e}$ describe how well $e = (v, u)$ is suited as a next-hop edge from node $v$ for packets destined for $z$. Producing such an output requires the network features to contain some form of positional embedding. However, by default, a GNN cannot spatially identify nodes and edges of the input topology in relation to each other due to its permutation equivariance property (Bronstein et al., 2021). Therefore, we provide auxiliary positional information as node input features by calculating APSP once, and then supplying the resulting path distances between nodes. For the path computation, we use the link weights calculated by EIGRP as described in Section B.5. For high-frequency RO, removing the need for APSP on every re-optimization reduces the overall computational complexity. Importantly, the actor module is still topology-agnostic because the topology and positional information is supplied as input, allowing *FieldLines* to generalize to novel topologies during inference.

The *selection* module $\psi : \mathcal{S} \times \mathbb{R}^{|V| \times |E|} \to \mathcal{A}$ treats the next-hop edge rating supplied by $\phi$ as logits over outgoing edges per node $v \in V$ and destination $z$. During training rollouts, $\psi$ uses Boltzmann exploration by sampling a next-hop edge from these distributions, using a learnable temparature parameter $\tau_\psi$. During evaluation, it simply chooses the maximum probability edges for each pair of routing node $v$ and destination node $z$. In both cases, the resulting next-hop choices correspond to rules in destination-based routing. Importantly, this policy design is able to form non-coalescent routing paths (i.e routing loops) because we do not force coalescence. Our results show that constraining our routing to coalescent paths is not necessary, and there is evidence that, in certain network situations, routing loops can even be beneficial to routing performance (Brundiers et al., 2021)[3].

## 6 Experiment Setup

In our experiments, we consider an episodic RL setting in which every training and evaluation episode comes with its own network scenario. This includes the network topology as well as traffic demands and link failure events for the entire length of simulation $T$. To account for packet-level dynamics in *PackeRL*, we simulate $H = 100$ steps per episode and set $\tau_{\text{sim}}$, the time simulated per environment step, to 5 ms, which yields $T = 500$ ms. This implies that, as opposed to existing fluid-based environments, training and evaluation in *PackeRL* involves a larger number of temporally fine-grained inference steps. Furthermore, our training and evaluation protocols cover varying network topologies, traffic situations, and the presence of link failures. Since such a setting requires routing algorithms to provide quick updates for arbitrary topologies without the need for re-training, we do not evaluate approaches that are hand-crafted or fine-tuned to a specific network topology. Instead, as Bernárdez et al. (2023) have shown that learning useful general-purpose representations for RO is possible, we design our evaluation to shed light on the importance of packet-level dynamics for learning more suitable representations.

---

[3]We are aware that the examples presented in (Brundiers et al., 2021) use Equal-Cost Multipath (ECMP) routing (Chiesa et al., 2016), which at the moment is not supported by *PackeRL*.

**FieldLines, Received (↑)**

| | low | medium | high | very high |
|---|---|---|---|---|
| UDP | -0.22%
(4.9MB) | -0.12%
(14.04MB) | +0.02%
(27.44MB) | +0.22%
(51.46MB) |
| TCP/UDP | -0.25%
(15.43MB) | +0.28%
(35.47MB) | +2.51%
(55.68MB) | +2.48%
(81.61MB) |
| TCP | -1.03%
(22.84MB) | +2.29%
(48.36MB) | +1.92%
(62.89MB) | +2.69%
(90.67MB) |

**FieldLines, Avg. Delay (↓)**

| | low | medium | high | very high |
|---|---|---|---|---|
| UDP | -1.54%
(7.24ms) | -1.99%
(7.15ms) | -2.03%
(7.3ms) | -2.44%
(8.27ms) |
| TCP/UDP | -2.43%
(6.78ms) | -2.24%
(7.04ms) | -2.82%
(7.31ms) | -2.35%
(7.86ms) |
| TCP | -1.88%
(6.83ms) | -2.37%
(7.21ms) | -1.84%
(7.39ms) | -1.46%
(7.48ms) |

**FieldLines, Dropped (↓)**

| | low | medium | high | very high |
|---|---|---|---|---|
| UDP | +0.05%
(0.08%) | +0.08%
(0.14%) | 0.0%
(0.3%) | -0.02%
(3.23%) |
| TCP/UDP | +0.15%
(0.57%) | +0.14%
(1.38%) | -0.05%
(3.0%) | -0.35%
(6.7%) |
| TCP | +0.17%
(0.98%) | -0.01%
(2.78%) | -0.14%
(5.61%) | -0.28%
(9.82%) |

**M-Slim, Received (↑)**

| | low | medium | high | very high |
|---|---|---|---|---|
| UDP | -0.4%
(4.89MB) | -0.19%
(14.03MB) | -0.11%
(27.41MB) | +0.85%
(51.78MB) |
| TCP/UDP | -2.36%
(15.11MB) | -0.37%
(35.24MB) | +3.61%
(56.28MB) | +4.58%
(83.28MB) |
| TCP | -2.34%
(22.54MB) | +3.57%
(48.97MB) | +3.25%
(63.71MB) | +4.1%
(91.91MB) |

**M-Slim, Avg. Delay (↓)**

| | low | medium | high | very high |
|---|---|---|---|---|
| UDP | -1.23%
(7.26ms) | -1.56%
(7.18ms) | -1.75%
(7.32ms) | -4.23%
(8.12ms) |
| TCP/UDP | -3.4%
(6.72ms) | -3.37%
(6.96ms) | -3.86%
(7.23ms) | -3.34%
(7.78ms) |
| TCP | -2.65%
(6.78ms) | -3.57%
(7.12ms) | -2.87%
(7.31ms) | -2.46%
(7.41ms) |

**M-Slim, Dropped (↓)**

| | low | medium | high | very high |
|---|---|---|---|---|
| UDP | +0.07%
(0.1%) | +0.13%
(0.2%) | +0.16%
(0.46%) | -0.56%
(2.69%) |
| TCP/UDP | +0.27%
(0.69%) | +0.15%
(1.4%) | -0.14%
(2.9%) | -0.58%
(6.48%) |
| TCP | +0.21%
(1.01%) | -0.09%
(2.7%) | -0.23%
(5.51%) | -0.54%
(9.57%) |

Figure 4: Results for our approaches *FieldLines* and *M-Slim* on the *nx–XS* topology preset, displayed for varying traffic kinds and intensities. Cells show the mean value over 100 episodes relative to EIGRP's performance in the first line, and the absolute mean value in the second line. Both approaches consistently improve the average packet delay. Moreover, for more intense traffic, they outperform EIGRP in goodput and drop ratio. The sending rate dynamics of TCP-dominated traffic amplify the reported difference.

## 6.1 Evaluated Routing Approaches

We compare MAGNNETO, which is trained in a fluid-based environment, to *M-Slim* and *FieldLines*, which are trained in *PackeRL*. To evaluate MAGNNETO in *PackeRL*, at each timestep, we feed it the latest TM of sent bytes. Beginning with zero LU and randomly initialized integer link weights, it uses current LU and link weights as edge features and increments one or more link weights. Its fluid-based environment then distributes the TM's traffic volumes across the flow network using the paths obtained from an APSP computation and obtains new LU values for the next iteration. This way, MAGNNETO uses the initial TM for $\Theta(E)$ optimization steps in which it iteratively adjusts the link weights for path computation. The final link weights are used to obtain the actual routing paths communicated to *PackeRL*. On the contrary, *M-Slim* and *FieldLines* are evaluated by doing one deterministic inference pass on the respective policy architecture, which directly yields shortest path weights in the case of *M-Slim*, and node-centric next-hop selections per packet destination in the case of *FieldLines*. In addition to the learned approaches, we consider two shortest-path baselines: OSPF is introduced in Section 3.1, and EIGRP (Savage et al., 2016) also involves link delays in its link weight calculation. These two routing protocols are widely adopted because their topology discovery mechanisms allow for routing in any network topology, and the heuristics used for computing shortest paths work well in most practical situations. Yet, by default, these protocols are oblivious to varying traffic conditions and network utilization. Therefore, for evaluating the two baselines it is sufficient to compute shortest paths once at the start of the episode and every time the network topology changes. We use standard reference values for the path computation process, which is explained in Section B.5. Lastly, we include results of two random policies for reference: *Random (NH)* chooses random next-hop edges for each possible destination $z \in V$ at each routing node $v \in V$, and *Random (LW)* uses shortest-path routing with randomly generated link weights. While *Random (LW)* corresponds to an untrained MAGNNETO/*M-Slim* policy, *Random (NH)* corresponds to an untrained *FieldLines* policy.

## 6.2 Network Scenarios

We consider five groups of randomly generated topologies of varying scale, generated as described in Appendix A.5. We call these groups the *nx* family and work with *nx–XS* (6–10 nodes), *nx–S* (11–25 nodes), *nx–M* (26–50 nodes), *nx–L* (51–100 nodes) and *nx–XL* (101–250 nodes). Figures 10 and 11 visualize example topologies. We train and evaluate on a grid of combinations of traffic scaling values $m_{\text{traffic}}$ and TCP fractions

**TCP/UDP, medium traffic, 11-25 nodes**

| | EIGRP | FieldLines (ours) | M-Slim (ours) | MAGNNETO |
|---|---|---|---|---|
| Action Fluctuation | 0% | +2.06% (1.67, 2.79) | +11.02% (9.27, 12.51) | +8.4% (3.64, 12.36) |
| Received (↑) | 65.55MB | -0.86% (63.66, 65.77) | +0.86% (65.14, 67.01) | -4.32% (59.54, 65.96) |
| Avg. Delay (↓) | 9.86ms | +0.19% (9.84, 9.92) | -0.73% (9.77, 9.82) | +2.28% (9.82, 10.49) |
| Dropped (↓) | 1.74% | +0.07% (1.76, 1.88) | -0.07% (1.56, 1.78) | +0.27% (1.66, 2.4) |

**TCP/UDP, medium traffic, 26-50 nodes**

| | EIGRP | FieldLines (ours) | M-Slim (ours) | MAGNNETO |
|---|---|---|---|---|
| Action Fluctuation | 0% | +1.16% (0.93, 1.44) | +12.32% (10.55, 13.83) | +9.83% (3.1, 14.57) |
| Received (↑) | 102.66MB | -0.16% (101.6, 103.32) | +3.15% (103.87, 107.43) | -3.61% (93.13, 105.8) |
| Avg. Delay (↓) | 12.63ms | +0.33% (12.61, 12.84) | -0.84% (12.49, 12.56) | +2.79% (12.56, 13.44) |
| Dropped (↓) | 1.93% | -0.02% (1.88, 1.94) | -0.29% (1.52, 1.83) | +0.22% (1.61, 2.69) |

**TCP/UDP, medium traffic, 101-250 nodes**

| | EIGRP | FieldLines (ours) | M-Slim (ours) | MAGNNETO |
|---|---|---|---|---|
| Action Fluctuation | 0% | +0.33% (0.24, 0.49) | +11.5% (10.56, 12.73) | N/A |
| Received (↑) | 221.72MB | -0.35% (220.53, 221.51) | +2.05% (224.68, 227.89) | N/A |
| Avg. Delay (↓) | 20.22ms | +0.11% (20.23, 20.27) | -0.57% (20.07, 20.13) | N/A |
| Dropped (↓) | 1.66% | +0.02% (1.66, 1.7) | -0.31% (1.29, 1.42) | N/A |

**TCP, high traffic, 11-25 nodes**

| | EIGRP | FieldLines (ours) | M-Slim (ours) | MAGNNETO |
|---|---|---|---|---|
| Action Fluctuation | 0% | +2.88% (2.02, 3.74) | +12.92% (11.73, 14.01) | +10.54% (5.73, 14.57) |
| Received (↑) | 117.17MB | +0.56% (115.96, 118.79) | +3.56% (117.85, 123.16) | -3.52% (107.42, 120.8) |
| Avg. Delay (↓) | 9.85ms | -2.6% (8.89, 9.84) | -0.61% (9.76, 9.8) | +1.09% (9.75, 10.18) |
| Dropped (↓) | 6.66% | -0.44% (5.24, 6.56) | -0.38% (6.07, 6.63) | +0.26% (6.31, 7.29) |

**TCP, high traffic, 26-50 nodes**

| | EIGRP | FieldLines (ours) | M-Slim (ours) | MAGNNETO |
|---|---|---|---|---|
| Action Fluctuation | 0% | +1.27% (1.02, 1.59) | +13.0% (11.74, 13.92) | +11.18% (3.8, 16.21) |
| Received (↑) | 185.16MB | -0.0% (182.0, 186.43) | +3.99% (188.27, 195.16) | -3.56% (167.91, 192.66) |
| Avg. Delay (↓) | 12.51ms | +0.43% (12.51, 12.73) | +0.21% (12.51, 12.55) | +1.66% (12.44, 12.95) |
| Dropped (↓) | 7.29% | -0.01% (7.21, 7.37) | -0.48% (6.66, 7.05) | +0.23% (6.82, 7.93) |

**TCP, high traffic, 101-250 nodes**

| | EIGRP | FieldLines (ours) | M-Slim (ours) | MAGNNETO |
|---|---|---|---|---|
| Action Fluctuation | 0% | +0.35% (0.22, 0.54) | +11.15% (10.97, 11.28) | N/A |
| Received (↑) | 372.96MB | +0.06% (372.86, 373.59) | +4.45% (380.87, 394.55) | N/A |
| Avg. Delay (↓) | 20.73ms | +0.03% (20.71, 20.78) | +0.1% (20.73, 20.77) | N/A |
| Dropped (↓) | 7.34% | +0.03% (7.3, 7.42) | -0.64% (6.6, 6.9) | N/A |

Figure 5: Results for the *nx–S* (11–25 nodes), *nx–M* (26–50 nodes) and *nx–XL* (101–250 nodes) presets. Cells show the mean values over 100 evaluation episodes (30 for *nx–XL*) in the first line, and min and max values across seeds in the second line. Values and colors are relative to EIGRP. Our approaches generalize to larger topologies, but the routing of *FieldLines* becomes more and more similar to that of EIGRP. We did not evaluate MAGNNETO on the *nx–XL* preset due to excessive inference times.

$p_{\mathrm{TCP}}$. Specifically, we use $m_{\mathrm{traffic}} \in \{0.25, 0.75, 1.5, 3.0\}$, which we denote as "low", "medium", "high", and "very high" traffic in the following, and $p_{\mathrm{TCP}} \in \{0\%, 50\%, 100\%\}$, which we denote as "UDP", "TCP/UDP" and "TCP". Finally, while we do not include link failure events in our experiments by default, in Section 7.2 we also provide an evaluation on the *nx–S* topology set that includes randomly generated link failures as described in Section A.6. This amounts to an average of 2.38 link failures per episode, distributed as shown in Figure 9 on the left side. Importantly, we do not work with pre-generated datasets of network scenarios. Instead, using *synnet* and controllable random seeds for training and evaluation, we generate a new network topology, traffic demands and link failures for all timesteps at the start of every episode.

### 6.3 Training and Evaluation Details

We train *M-Slim* and *FieldLines* on 16 random seeds for 100 iterations of 16 episodes each. We train *FieldLines* on the *nx–XS* topologies while restricting the training of *M-Slim* to the two well-known network topologies NSFNet and GEANT2, with link datarate values taken from Bernárdez et al. (2023), for better comparability to MAGNNETO. For both approaches, we use PPO (Schulman et al., 2017) and refer to Section B.7 for hyperparameter details. Given the episode length $H = 100$, this results in a total of $160\,000$ training steps per training run which, depending on TCP ratio and amount of traffic, take between 3 and 14 hours of training on 4 cores of an Intel Xeon Gold 6230 CPU. For *FieldLines*, the first five iterations are warm-start iterations. In these iterations, we replace the actions sampled during rollout by the ones suggested by the EIGRP baseline. We train MAGNNETO on 8 random link weight initialization seeds as per their protocol (Bernárdez et al., 2023), using their fluid-based environment and PPO implementation. The performance values presented in this work are obtained by taking the mean over 100 evaluation episodes, except for *nx–XL* for which we use 30 evaluation episodes. For the learned approaches, we exclude non-convergent runs by showing the performance of the better-performing half of the used random seeds.

## 7 Results

We describe the results of our experiments in Sections 7.1 to 7.3 and discuss their implications in Section 7.4.

| | TCP/UDP, medium traffic | | | | TCP, high traffic | | | | TCP, very high traffic | | | |
| | EIGRP | FieldLines (ours) | M-Slim (ours) | MAGNNETO | EIGRP | FieldLines (ours) | M-Slim (ours) | MAGNNETO | EIGRP | FieldLines (ours) | M-Slim (ours) | MAGNNETO |
|---|---|---|---|---|---|---|---|---|---|---|---|---|
| Received (↑) | 62.82MB | -0.71% (61.91, 62.77) | +1.21% (62.77, 64.38) | -4.36% (55.95, 63.35) | 106.06MB | -0.04% (103.65, 107.19) | +2.66% (106.44, 110.43) | -4.32% (95.47, 108.3) | 142.29MB | +0.67% (142.12, 143.7) | +2.76% (144.72, 147.43) | -3.55% (129.93, 145.96) |
| Avg. Delay (↓) | 9.94ms | -0.0% (9.91, 9.97) | -0.94% (9.82, 9.87) | +2.08% (9.85, 10.59) | 10.23ms | -3.45% (8.93, 10.2) | -0.63% (10.15, 10.17) | +0.74% (10.1, 10.51) | 9.99ms | -2.61% (8.98, 9.98) | -0.06% (9.97, 10.0) | +0.28% (9.84, 10.15) |
| Dropped (↓) | 2.14% | +0.07% (2.16, 2.24) | -0.11% (1.89, 2.21) | +0.3% (2.05, 2.97) | 7.42% | -0.52% (5.66, 7.34) | -0.36% (6.88, 7.34) | +0.29% (7.15, 8.12) | 12.23% | -0.59% (10.18, 12.15) | -0.33% (11.83, 12.03) | +0.22% (11.84, 12.82) |

Figure 6: Results on the *nx–S* topology preset with link failures. Cells show the mean values over 100 evaluation episodes in the first line, and min and max values across random seeds in the second line. Values and colors are relative to EIGRP. While all learned approaches can adapt to link failures, only our approaches stay above EIGRP. When dealing with link failures, *FieldLines* favors lower-delay paths over higher goodput.

## 7.1 Learning to Route in *PackeRL*

In Figure 3, we report results of all evaluated approaches on the *nx–XS* topology preset. EIGRP slightly outperforms OSPF in all evaluated scenarios, which is the reason why we do not include OSPF in the remaining results. Furthermore, the two random policies perform significantly worse than all other approaches. Concerning the learned approaches, for MAGNNETO, unlike the results reported in Bernárdez et al. (2023), even the best run does not beat static shortest-paths routing. On the other hand, our policies *M-Slim* and *FieldLines* provide routing performance that rivals EIGRP, with consistently lower average packet delays. Also, as further illustrated by Figure 4, both our policy designs improve goodput and packet drop ratio for network scenarios with high-intensity and TCP-dominated traffic. Unlike Figure 3, each matrix of Figure 4 displays the values of a single metric and a single approach, but over a grid of traffic setups of varying intensity and TCP ratio. This way, it becomes evident that the performance *FieldLines* and *M-Slim* relative to EIGRP improves for scenarios with intense and TCP-heavy traffic, and that learned high-frequency RO is particularly beneficial for such kinds of traffic. On the other hand, for scenarios with low traffic intensity, EIGRP is able to maintain a slightly higher goodput.

## 7.2 Generalizing to Unseen Network Topologies

Figure 5 shows the results of evaluating above section's policies on the larger topology presets *nx–S*, *nx–M*, and *nx–XL*. Relative to EIGRP, MAGNNETO's performance stays inferior yet stable, but due to the excessive inference times depicted in Section 7.3, we did not evaluate it on topology presets larger than *nx–M*. Interestingly, *M-Slim* is able to increase its goodput advantage over EIGRP with increasing network scale while maintaining solid values for other metrics. On the other hand, *FieldLines*'s performance becomes more and more similar to that of the EIGRP baseline as the network scale increases.

New network topologies can also arise due to failure events. Figure 6 shows results on the *nx–S* topology preset with randomly generated link failures as described in Section A.6. All learned approaches are able to adapt in the face of one or more link failures. However, *FieldLines* and *M-Slim* seem to behave slightly differently upon registering a link failure: While *FieldLines* favorizes a lower average packet delay, *M-Slim* maintains goodput and instead gives away its delay advantage. In any case, both our policy designs drop less data after link failures in traffic-intense situations.

## 7.3 Simulation and Inference Speed

The left side of Figure 7 reports the mean inference time of the evaluated policies on the various network scales of the *nx* topology preset. MAGNNETO's optimization loop quickly leads to inference times of several seconds. The single-pass inference of *M-Slim* and *FieldLines* brings down inference time considerably, but for large networks the inference time of *M-Slim* exceeds one second because it requires an APSP pass every step. *FieldLines*, which only requires that when the topology changes, keeps its inference times in the millisecond

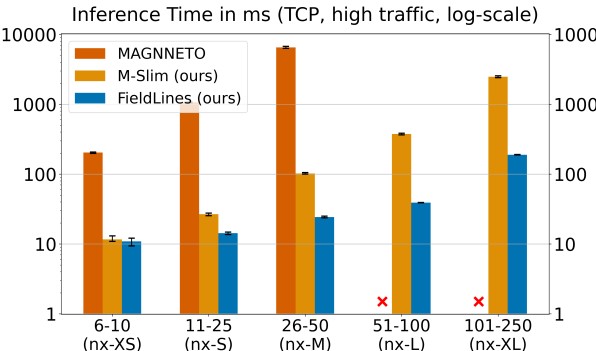 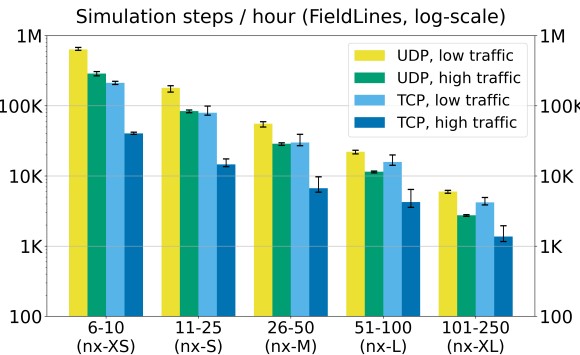

Figure 7: Left side: Mean inference times per *PackeRL* step on different network sizes. Our policies reduce the inference time required by MAGNNETO, which was not evaluated on larger topologies due to excessive inference times, by multiple orders of magnitude. Right side: Simulation steps per hour, including shared memory communication but excluding inference and learning times. Simulating TCP traffic in *PackeRL* is more costly than UDP traffic, and simulation speed depends on traffic intensity and network scale.

range for all evaluated graph scales. The right side of Figure 7 shows that simulating TCP traffic in *PackeRL* is more costly than UDP traffic, and that simulation speed depends on traffic intensity and network scale.

## 7.4 Discussion

The results shown in earlier parts of this section suggest several findings, which we will discuss in the following.

Firstly, the two random policies shown in Figure 3 perform notably worse than all other approaches. The performance of *Random (NH)* in particular is far off, since it does not check for routing path coalescence and thus frequently includes routing loops and detours in its actions. This shows that obtaining high-quality routing actions is not a trivial task, and that using RL to learn to route is beneficial. Next, we note a contrast between the performance of MAGNNETO and *M-Slim*. While MAGNNETO performs worse than the static shortest-path baselines (EIGRP), *M-Slim* outperforms them in all scenarios except those with low-volume traffic. This shows that placing a routing policy trained in a fluid-based environment into a packet-based one hurts performance, which is avoidable by directly training in a packet-based environment like *PackeRL*. In general, both our RL-powered approaches outperform EIGRP in scenarios with more intense traffic. We assume that congestion events happen more frequently in these scenarios and that the dynamic routing provided by *M-Slim* and *FieldLines* can better deal with this challenge. We also note that the advantage over EIGRP is slightly more pronounced for TCP-heavy traffic. This may be explained by TCP's sending rate adjustment. TCP actively probes for the maximum sending rate maintainable without loss of data. This likely leads to more congestion events, which in turn may be handled better by an adaptive routing policy.

Section 7.2 suggests that all learned approaches can deal with link failures effortlessly and that they generalize to large topologies even though they have been trained on much smaller and/or random topologies. Interestingly, while *M-Slim* seems to maintain its relative advantage over EIGRP, *FieldLines* seems to prioritize maintaining low latency over maximum global goodput when faced with link failures. Also, we note that the advantage of *FieldLines* over EIGRP disappears with growing network topology size. A possible explanation for the generalization behavior of *FieldLines* is indicated by the "Action Fluctuation" row of Figure 5. It denotes the average percentage of next-hop decisions over all routing and destination nodes rate that change between a timestep and the next. Evidently, the action fluctuation is much lower for *FieldLines* than for *M-Slim*, which indicates that *FieldLines* changes its routing actions much less frequently. Figure 21 of the appendix provides an illustrating example for this difference. Moreover, as the network scale increases, the routing of *FieldLines* becomes even more static. We suspect that this is the reason why, with growing network size, *FieldLines*' performance becomes more and more similar to that of EIGRP. The performance reported for *M-Slim* in Figures 5 and 6 suggests that re-optimizing routing less conservatively may further increase the performance of *FieldLines*. We hypothesize that, in order to increase the dynamicity of *FieldLines*

for large network topologies, we will need to adjust the action selection mechanism used during training and/or evaluation. For now, given the larger inference times of *M-Slim* on large network topologies, choosing between *M-Slim* or *FieldLines* is currently a matter of trading routing performance for responsiveness.

## 8 Conclusion

This work highlights the importance of packet-level simulation environments for RL-enabled routing optimization, together with policy designs that are trainable in such environments. Firstly, we show that a recently proposed approach trained in a fluid-based network environment fails to reproduce its training performance in packet-based simulation. Adapting the approach to the more realistic packet-based simulation setting negates the decline in performance and shows why packet-based training and evaluation environments are even needed. Our adaptation called *M-Slim* also cuts the time needed to re-optimize routing by one to two orders of magnitude, but still requires more than a second for large network topologies. Our novel next-hop selection policy design *FieldLines* is the first of its kind that can optimize routing within milliseconds for any network topology without the need for re-training. Finally, our evaluation setup demonstrates the versatility of our new packet-level training and evaluation framework *PackeRL*. It supports training and evaluating policies on a wide range of realistic network topologies and traffic setups, provides closed-loop simulation via its *ns-3* network simulator backend, and facilitates a detailed analysis of RL-based approaches with respect to a handful of well-known performance metrics. With the findings of this work, we hope to inspire future research on RL-enabled routing optimization, and provide a few pointers in the following final subsection.

### 8.1 Future Work

We have demonstrated the usefulness of *PackeRL* only for single-path routing but not for multi-path routing. This is due to the absence of standard Multipath TCP (MPTCP) implementations for *ns-3* or any other popular packet-level network simulator[4]. As existing implementations of MPTCP for *ns-3* are incomplete with respect to the official specification (Nadeem & Jadoon, 2019; Ford et al., 2020), we leave the extension of *PackeRL* to multipath routing for future work.

In this work, for simplicity, we have evaluated the routing performance when optimizing for a single objective or a weighted sum of multiple components with fixed weights. But in different network scenarios, the relative importance of these metrics may vary. Further efforts may investigate the relative importance of the optimization components in different network scenarios and turn to Multi-Objective RL (Hayes et al., 2022) to train a family of policies that provides better control over performance.

Despite the strong results of *FieldLines*, Section 7.4 has shown that its routing becomes increasingly static as network sizes grow, making it gradually lose its advantage. Figure 16 suggests that adding more message passing steps to the MPN module does not solve the problem on its own, but revisiting the action-selection mechanism used during the training of *FieldLines* may alleviate this issue. Furthermore, despite greatly reduced inference times, *FieldLines* still requires several ms to re-optimize routing. This does not influence routing in our experiments because we pause the environment during action selection as is commonly done in RL research. Long inference times may however degrade routing quality in the even more realistic setting where network operation would continue during action inference. Learning sub-second routing optimization with RL in such a soft real-time setting (Marchand et al., 2004) is an interesting and challenging avenue for further research, not least because hardware specifications will start to influence the routing results.

Finally, we note that the RL approaches considered in our experiments were trained and evaluated in a fully centralized manner. This raises the question: How do we account for the delays incurred by communicating state information across the computer network? High-frequency RO policies may need to respect Age of Information (AoI) (Yates et al., 2021) when capturing and processing the network state. Here, redistributing routing control to the nodes could reduce the influence of AoI, as nodes may restrict the topological coverage of input and output to their neighborhoods. It is an open question whether such systems of networked agents can achieve efficient routing, and if so, how they should be designed and deployed. We believe that building on *FieldLines*' next-hop routing design can help to answer this question in future work.

---

[4]Consequently, other recent RO approaches like TEAL (Xu et al., 2023) can not yet be evaluated in packet-level simulation.

**Broader Impact Statement**

Automating computer networks promises to greatly increase operational efficiency and save costs through over-provisioning or manual configuration. Here, RL-powered RO may become a cornerstone in autonomous computer networks. Other infrastructures like road networks or power grids may also benefit from this progress, given their structural similarity. Our work opens the door for future research on automation for such kinds of systems. Anyhow, as for most RL application domains, misusing RL approaches in computer networks for malicious intents is conceivable. Specifically, the black-box nature of RL-powered RO approaches can be abused to infiltrate the network's decision making, causing disturbances or loss of data if appropriate security measures are not taken. Furthermore, learned routing approaches may put certain kinds of traffic at an unnatural disadvantage in order to optimize overall routing performance. In addition to mindful deployment by the network engineers, RL-powered networking components should therefore include safeguards against the most common attack vectors.

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

# A  *PackeRL*: Framework Details

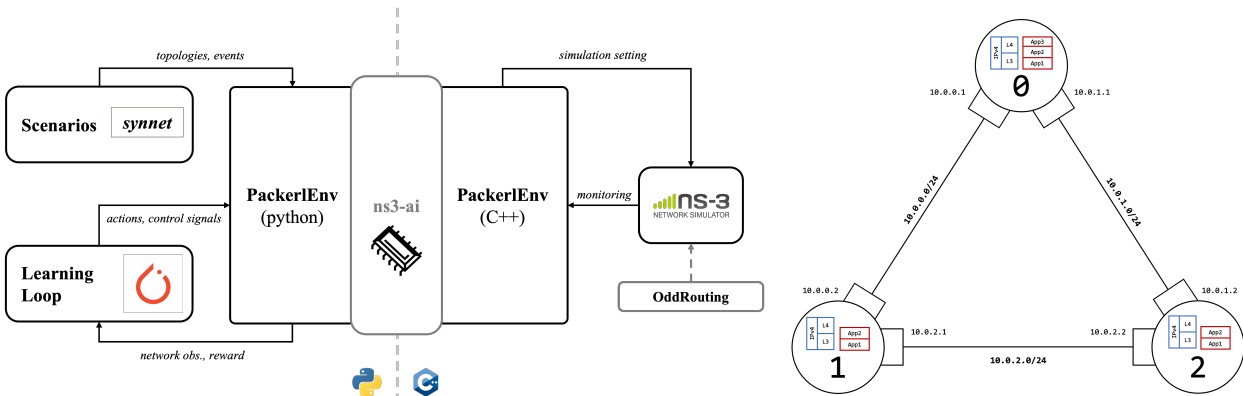

Figure 8: Left: Structural overview of *PackeRL*. Right: Example 3-node network setup in ns-3 incl. applications (red boxes) and Internet Stack (blue boxes).

## A.1  Network Structure and Simulation in ns-3

Networks in *ns-3*, by default, consist of nodes and links/connections between nodes, as illustrated in Figure 8. For modeling simplicity, we limit ourselves to connected network topologies that hold full-duplex Point-to-Point (P2P) connections transmitting data error-free and at a constant pre-specified datarate. Nodes themselves do not generate or consume data; Instead, applications are installed on nodes that generate data destined for other applications, or consume the data that is destined for them (red boxes in example network nodes in Figure 8). To transport data between nodes we install an *Internet Stack* on top of each node, adding IP and TCP/UDP components in a way that mimics the OSI reference model (Zimmermann, 1980). Also, nodes do not put data on the P2P link themselves, or read data from it. This is done by the network devices (rectangles attached to the nodes in Figure 8) that belong to a P2P connection, which are installed as interfaces on the two nodes that are being connected. Upon installation of the Internet Stack, the P2P connection between two nodes is assigned an IPv4 address space, with concrete IPv4 addresses given to the incident network devices.

## A.2  *PackeRL*'s Online Interaction Loop

*PackeRL* uses a two-component Python/C++ environment, with inter-component communication realized by shared memory module of *ns3-ai*. The Python component provides a Gymnasium-like interface to the learning loop, while the C++ component is a wrapper and entry point for simulations in *ns-3*. Initially, the Python environment starts an instance of its C++ counterpart in a subprocess, providing general simulation parameters (c.f. Section B.1) and a network scenario generated with *synnet*. The C++ environment enters its simulation loop and first installs the network topology in *ns-3*: It configures nodes, links and network devices accordingly, as well as a TCP and a UDP sink application per node. Then, it starts the *ns-3* `Simulator` that runs the actual simulation steps. The initial network state $S_0$ is communicated from the C++ component to the Python environment. Within the episode loop, the Python part provides the current network state $S_t$ to learned or baseline policies, and communicates the routing action $\mathbf{a}_t$ to the C++ component alongside the upcoming traffic and link failure events. Source applications are then created according to the upcoming traffic demands, with sending start times set to the respective demand arrival times. For TCP traffic demands, the source application attempts to send its data as quickly as possible, and we use *ns-3*'s default TCP CUBIC (Ha et al., 2008) to modulate the actual sending rate. Note that we currently do not support TCP Selective Acknowledgement (SACK) (Mathis et al., 1996) due to a presumed bug in *ns-3* that causes simulation crashes. For UDP traffic demands, we use the sending rate provided by the scenario as explained in Section A.6. The provided routing actions are installed as described in Section A.3. Next, using the `Simulator`, *ns-3* simulates the installed network for a duration of $\tau_{\mathrm{sim}}$: The installed source applications

(one per traffic demand) send data to the specified destination nodes as configured, which gets wrapped into IP packets as they enter the routing plane. The RP that has been installed with the Internet Stack fills each node's routing table and performs lookups when outgoing or incoming IP packets arrive, forwarding these packets to the specified next-hop neighbor or locally delivering them to the sink applications. Each node has a TCP and a UDP packet sink. After having simulated for a duration of $\tau_{\text{sim}}$, the C++ component pauses the simulation and obtains the network state $S_{t+1}$ for the completed timestep $t$. It communicates $S_{t+1}$ to the Python component that uses it to obtain $r_t$. After $H$ timesteps, the episode is done and the Python environment component sends a `done` signal to the C++ subprocess, which in turn ends its simulation loop and concludes the subprocess.

### A.3  Installing and Using Routing Actions in *ns-3*

Routing in computer network involves two primary tasks: determining the best paths and forwarding the packets along these paths. For the latter, routers keep a set of routing rules in their memory. Each rule specifies a next-hop neighbor to which packets with a certain destination are to be forwarded. This rule set - also known as a routing table - gets populated by the installed RP, and we adopt this mechanism for our work. Most contemporary IP networks employ destination-based routing, i.e. routers forward packets by finding the routing table entry that matches the packet's destination address, and sending the packet to the next-hop neighbor specified in that entry. As mentioned in Section 3, we adopt the commonly used forwarding mechanism using routing tables stored in the router's memory that may get updated by the RP. Therefore, regardless of the routing action's representation, we need to convert it into a set of routing table rules for each concerned routing node, where each rule contains a next-hop neighbor preference for a given destination node. We do the conversion prior to placing the actions into the shared memory module on the Python side. On the C++ side, the `OddRouting` module serves as a drop-in replacement for other routing protocols like OSPF that allows the installation of the provided routing next-hop preferences onto the network nodes. Since not all node pairs in a network necessarily communicate with each other, the `OddRouting` module stores the received routing preferences in a separate location on the routing node, and only fills the nodes' routing tables on-demand once a packet arrives at a node for which no suitable routing rule exists in its routing table. All subsequent packets destined for the same target node will have access to the newly installed table entry until the start of a new timestep, when new routing preferences will be stored in the node and its routing table will be flushed. Otherwise, `OddRouting` resembles the other IPv4 RPs implemented in *ns-3*, leveraging the line-speed capability of the forwarding plane.

### A.4  Network Monitoring in *ns-3*

In order to efficiently obtain the state of the computer network, we utilize the In-Band Network Telemetry capabilities provided by *ns-3*'s `FlowMonitor` module (Carneiro et al., 2009). It utilizes probes installed on packets to track per-flow statistics such as traffic volume, average and maximum packet delay, and node-level routing events down to the IP level. We use this information to obtain TMs of packets/bytes sent and received (for visualization and the MAGNNETO baseline), as well as global average/maximum packet delay and node-level traffic statistics that form part of the monitored network state $S$. Moreover, our network topology is modeled as a graph with edges that consist of physical connections between network devices installed on nodes. `FlowMonitor` does not capture queueing and drop events happening on the network device level, and we therefore also report events happening in network devices and channels to obtain edge-level information on link utilization, packet buffer fill, and bytes/packets sent/received/dropped. Since the P2P connections are full-duplex, we model the network monitoring as a directed graph where an edge of the original network topology is replaced with one edge in each direction. These directed edges contain the state of the respective sender device, i.e. edge $(u, v)$ contains packet buffer load, link utilization and traffic statistics for traffic buffered in $u$ flowing to $v$. At the end of a timestep $t$, $S_t$ holds global, node and edge features that reflect the overall network performance and utilization during timestep $t$, as well as its load state at the end of timestep $t$. For the edges, we add their datarate, delay and packet buffer capacities to the list of features. For the initial state $S_0$, all utilization and traffic values are set to zero.

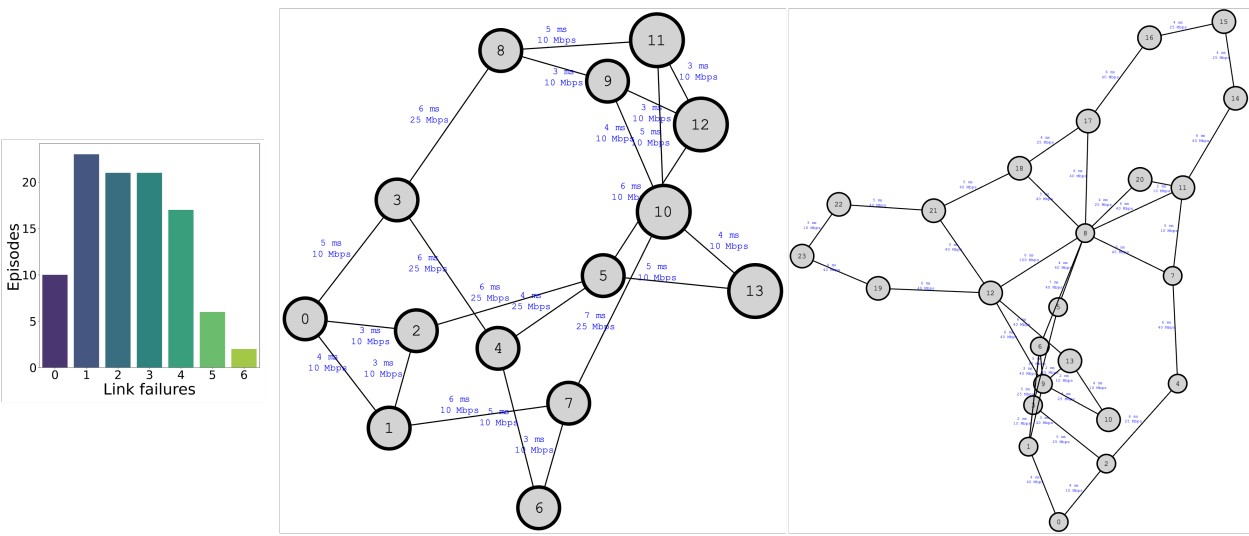

Figure 9: Visualizations of graph scenarios. Left: Link failure distribution across episodes for *nx-s* with link failures. Center: NSFNet as used by MAGNNETO. Right: GEANT2 as used by MAGNNETO. Note that node positions for visualization are not provided by MAGNNETO for NSFNet2 and GEANT2, and therefore the visualizations shown here may differ from related work.

### A.5  *synnet*: Generating Network Topologies

Network topologies vary greatly depending on the scope and use case of the network. For this work, we orientate our scenario generation towards the topologies spanned by the edge routers that connect datacenters in typical Inter-Datacenter Wide Area Networks (Inter-DC WANs). These are usually characterized by loosely meshed powerful edge routers and high-datarate medium-latency links that connect two such routers each. While we also employ link delay values in the low ms range, we scale down typical datarate values for Inter-DC WANs to lie in the high Mbps range, to speed up simulation times under stress situations without loss of generality of the simulation results. For simplicity, we set the packet buffer sizes of network devices incident to P2P connections to the product of link datarate and round-trip delay, which is common throughout the networking literature (Spang et al., 2022).

To generate random network topology graphs, we use the ER model (Erdős et al., 1960) and the WS model (Watts & Strogatz, 1998), and, up to 50 nodes, the BA (Barabási, 2009) model. All models are available via NetworkX graph analysis package (Hagberg et al., 2008). Figures 10 and 11 show examples for such random topology graphs. In any case, nodes and edges are assigned unique integer IDs for identification purposes. To add the missing datarate and delay values to the links, we follow the following steps:

1. We first embed the random graph into a two-dimensional plane using the Fruchterman-Reingold force-directed algorithm (Fruchterman & Reingold, 1991) to create synthetic positional information for the random graph's nodes, similar to the position information provided for nodes in related network datasets (Orlowski et al., 2010; Knight et al., 2011; Spring et al., 2002).

2. The resulting positional layout is centered around the two-dimensional point of origin, which we use to obtain location weights per node that are inversely proportional to its distance to the origin. We scale the location weights $\mathbf{w}_p$ to lie in $[1, w_{p_{\max}}]$.

3. We obtain degree weights $\mathbf{w}_d$ from the array of node degrees scaled to lie in $[1, w_{d_{\max}}]$.

4. We obtain node traffic potentials $\mathbf{c}' = \lambda_c \mathbf{w}_d + (1 - \lambda_c) \mathbf{w}_p$ by using a weight tradeoff parameter $\lambda_c$, and normalize them by dividing by the max value to obtain normalized node traffic potentials $\mathbf{c}$.

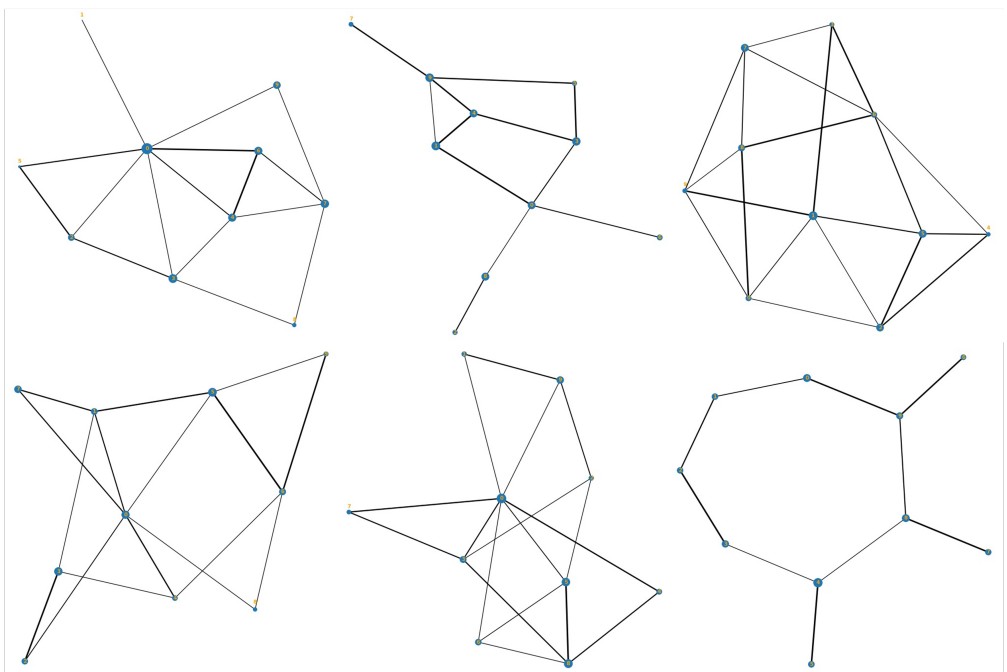

Figure 10: Examples of 10-node network topologies generated with NetworkX. Bigger nodes indicate higher node weights, thicker edges indicate higher edge weights. Columns from left to right (2 examples each): Barabási-Albert (BA), Erdős-Rényi (ER), Watts-Strogatz (WS).

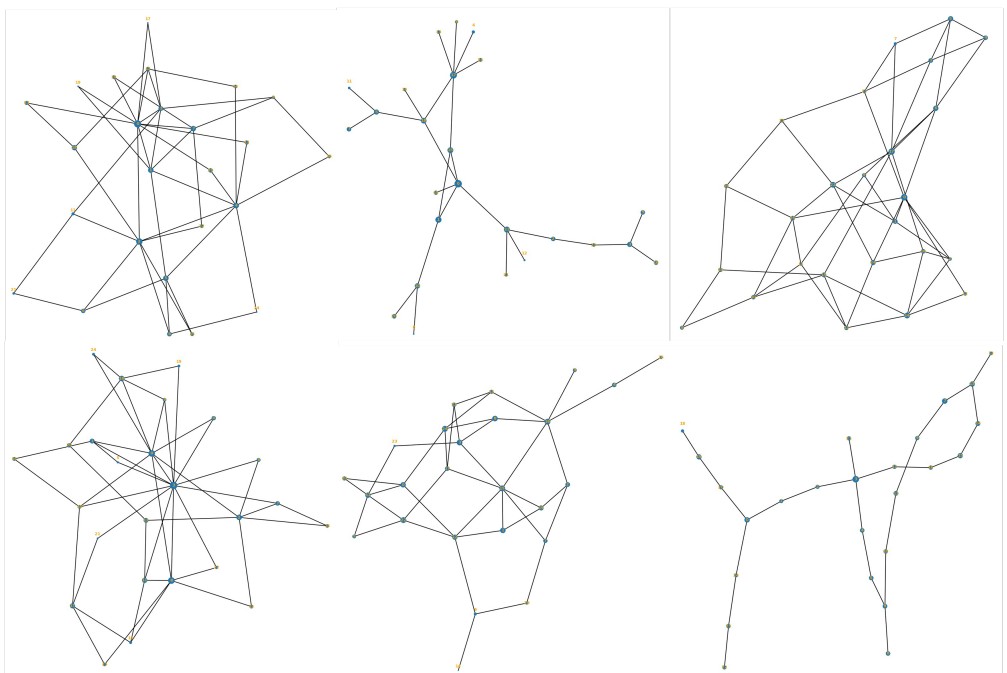

Figure 11: Examples of 25-node network topologies generated with NetworkX. Bigger nodes indicate higher node weights, thicker edges indicate higher edge weights. Columns from left to right (2 examples each): BA, ER, WS.

5. The edge delay values are obtained by calculating the euclidean distance between the adjacent nodes, randomly perturbing them by $\delta_{\mathrm{rand}}$ and rescaling them to average $m_{\mathrm{delay}}$ with a minimum delay value of 1 ms.

6. The datarate values per edge $(i, j)$ are obtained by taking the greater of the incident nodes' traffic potentials $c_i$ and $c_j$, randomly perturbing it by $\delta_{\mathrm{rand}}$ and rescaling it to lie in the pre-specified interval of minimum and maximum datarates $[v_{\mathrm{min}}, v_{\mathrm{max}}]$.

### A.6  *synnet*: Generating Traffic and Link Failure Events

Our process to generate flow-level traffic is inspired by reported traffic characteristics of real-world data centers (Benson et al., 2010):

1. Inspired by gravity TMs (Roughan, 2005), we generate a "traffic potential matrix" $\mathbf{B} = \mathbf{c}\mathbf{c}^T$ and randomly perturb its values by $\delta_{\mathrm{rand}}$. Its values $b_{ij}$ describe the *expected* (not actually measured) relative traffic intensity between each source-destination node pair $i, j$ and will be used for the upcoming demand generation. Diagonal entries are set to 0 to exclude self-traffic, meaning that traffic demand generation is skipped.

2. We use a flow size tradeoff parameter $\lambda_{\mathrm{flow}} \in [0, 1]$ to balance the frequency and size of arriving flows. Smaller values lead to more but smaller traffic demands, larger values lead to fewer but larger traffic demands.

3. In order to sample inter-arrival times between demands for each pair of nodes $i, j$, we use a log-logistic distribution with a fixed shape parameter $\beta_{\mathrm{t}} > 0$ and a scale $\alpha_{\mathrm{t}} = \frac{\lambda_{\mathrm{flow}} + \frac{1}{5}}{m_{\mathrm{traffic}}}$ that uses a traffic scaling parameter $m_{\mathrm{traffic}}$ depending on the evaluated task and class of graph topology. Starting from $\tau = 0$, we sample inter-arrival times for each node pair until we have reached the total simulated time scaled by traffic intensity $\tau = b_{ij} H \tau_{\mathrm{sim}}$ (with $H$ being the episode length, $\tau_{\mathrm{sim}}$ being the simulated time per episode step, and arrival times obtained via cumulative summation of inter-arrival times). Finally, we obtain the actual demand arrival times $\tau < H\tau_{\mathrm{sim}}$ per node pair $i, j$ by dividing the generated arrival times by $b_{i,j}$, capping at 50 ms.

4. For each generated traffic demand, we sample a demand size using a Pareto distribution. It uses a fixed scale parameter $\alpha_{\mathrm{s}}$ that also specifies the minimum demand size in bytes, and a shape $\beta_{\mathrm{s}} = \beta_{s_{\mathrm{base}}} + \log(\lambda_{\mathrm{flow}}^{-\frac{1}{37}})$ depending on a shape base parameter $\beta_{s_{\mathrm{base}}}$ that determines the tail weight of the demand size distribution. We cap the demand sizes at 1 TB.

5. A fraction $p_{\mathrm{TCP}} \in [0, 1]$ of the generated traffic demands is marked as TCP traffic demands, with the rest being marked as UDP demands. While the simulation will try to finish TCP demands as quickly as possible and under the sending rate moderation of TCP, we assign a constant sending rate of 1 Gbps for UDP demands of less than 100 KB, and a constant sending rate drawn uniformly from $[1, 5]$ Mbps for all other UDP demands.

For creating link failure events, for each step of the episode, we obtain link failure probabilities from a Weibull distribution as explained in Section B.3 and draw a boolean sample for each edge that determines whether it will fail in the upcoming step. In order to maintain the connected-ness of the network topology, we then restrict the creation of link failure events to edges that are non-cut at the time of event.

# B    Hyperparameters, Configuration And Defaults

The listed default hyperparameters and settings are used in all our experiments unless mentioned otherwise.

## B.1    Simulation in *ns-3*

We set up the applications to send data packets of up to 1472 bytes, which accounts for the commonly used IP packet maximum transmission unit of 1500 bytes and the sizes for the IP (20 bytes) and ICMP (8 bytes) packet header. UDP packets thus are 1500 bytes large, whereas TCP may split up data units received from the upper layer as required. We set the simulation step duration $\tau_{sim}$ to 5 ms and make each episode last $H = 100$ steps, simulating a total of $T = 500$ ms per episode.

## B.2    *synnet*: Topology Generation

For the BA model we use an attachment count of 2, and stop using the BA model altogether for networks above 50 nodes as kurtosis of the node degree distribution becomes too high at that point. For the ER model we set the average node degree to 3, and for the WS we choose a rewiring probability of 30% and an attachment count of 4. For a deterministic positional embedding of the graphs' nodes via the Fruchterman-Reingold force-directed algorithm, we set its random seed to 9001. We set the maximum node location weight $w_{p_{\max}}$ and the maximum node degree weight $w_{d_{\max}}$ to 10, and use a weight tradeoff parameter of $\lambda_c = 0.6$. We use a base average edge delay value $m_{\mathrm{delay}}$ of 5 ms, minimum and maximum edge datarate values of $v_{\min} = 50e6$ and $v_{\max} = 200e6$, and a random perturbation of $\delta_{\mathrm{rand}} = 0.1$ (i.e. random perturbation by up to $\pm 10\%$).

## B.3    *synnet*: Traffic and Link Failure Event Generation

For random perturbation, we again use $\delta_{\mathrm{rand}} = 0.1$ (i.e. random perturbation by up to $\pm 10\%$). The flow size tradeoff parameter $\lambda_{\mathrm{flow}}$ is set to 0.5, the demand interarrival time distribution shape parameter $\beta_t$ is set to 1.5, the demand size distribution scale parameter $\alpha_s$ is set to 10 (i.e. demands are at least 10 bytes), and the distribution shape base parameter is set to $\beta_{s_{\mathrm{base}}} = 0.4$. As per Bogle et al. (2019), we model link failure probabilities per simulated step as a Weibull distribution $W(\lambda, k)$, using a shape parameter $\lambda = 0.8$ and a scale parameter $k = 0.001$. Figure 12 illustrates the resulting probability distributions for traffic demands and link failures.

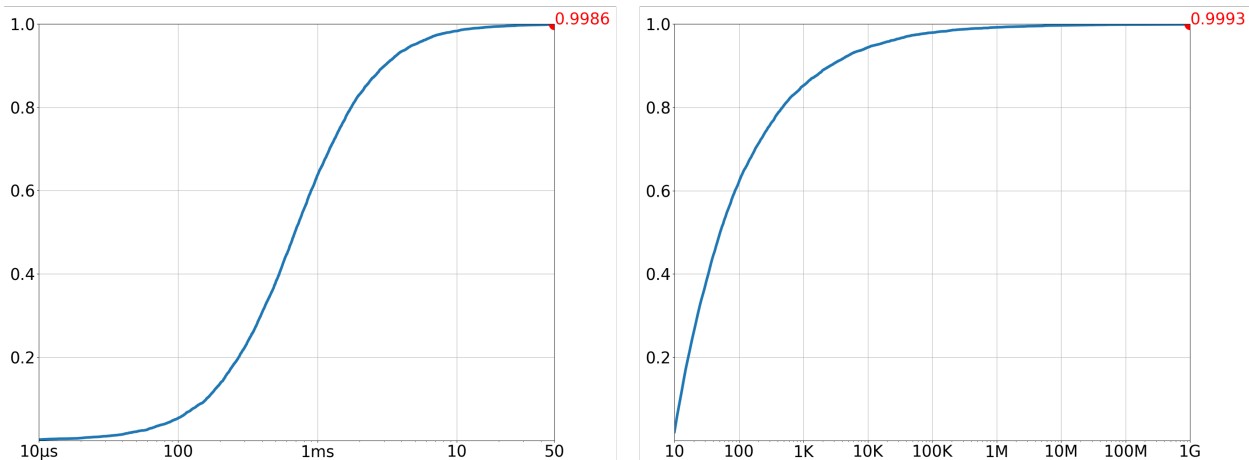

Figure 12: Cumulative probabilities for demand interarrival times (left, log-logistic distribution) and demand sizes in bytes (right, Pareto distribution). The red points at the end of the curves denote the cumulative probability at 50 ms and 1 GB.

### B.4 Monitoring Features

The following features are monitored in our *ns-3* simulation:

- **global:** maximum link utilization ($\texttt{maxLU} \in [0,1]$), average datarate utilization $\texttt{avgTDU} \in [0,1]$, average packet delay $\texttt{avgPacketDelay} \in \mathbb{R}^+$, maximum packet delay $\texttt{maxPacketDelay} \in \mathbb{R}^+$, average packet jitter $\texttt{avgPacketJitter} \in \mathbb{R}^+$, globally sent/received/dropped/retransmitted bytes in $\mathbb{N}_0$.

- **edge:** link utilization $\texttt{LU} \in [0,1]$, maximum relative packet buffer fill $\texttt{txQueueMaxLoad} \in [0,1]$, relative packet buffer fill at end of simulation step $\texttt{txQueueLastLoad} \in [0,1]$, packet buffer capacity in $\mathbb{N}^+$, channel datarate and delay in $\mathbb{N}^+$, sent/received/dropped bytes in $\mathbb{N}_0$.

- **node:** sent/received/retransmitted bytes in $\mathbb{N}_0$.

By default, we do not use the node features in our experiments. This is because our feature ablation in Section C.5 shows that including this information globally and on an edge level is enough and leads to the best performance. Consequently, for our experiments we have $d_U = 9$, $d_E = 10$ and $d_V = 0$ for the monitored network state $G_t$. Also, we normalize all input features akin to Schulman et al. (2017). Finally, we stack the four latest observations $(S_{t-3}, S_{t-2}, S_{t-1}, S_t)$, using zero-value padding.

### B.5 OSPF and EIGRP Weight Calculation

The default calculation formula for OSPF link weights is

$$\text{weight}(e) = \frac{v_{\text{ref}}^{\text{OSPF}}}{v(e)}$$

where $v(e)$ denotes the datarate value of link $e$ and the reference datarate value $v_{\text{ref}}^{\text{OSPF}}$ is set to $10^8$ (Moy, 1997). For EIGRP link weights, we use the classic formulation with default K-values, which yields

$$\text{weight}(e) = 256 \cdot \left( \frac{v_{\text{ref}}^{\text{EIGRP}}}{v(e)} + \frac{d(e)}{d_{\text{ref}}^{\text{EIGRP}}} \right)$$

where $d(e)$ denote the delay value of link $e$, the reference datarate value $v_{\text{ref}}^{\text{EIGRP}}$ is set to $10^7$ and the reference delay value $d_{\text{ref}}^{\text{EIGRP}}$ is set to 10 (Savage et al., 2016). The two routing protocols use the link weights to compute routing paths using the Dijkstra algorithm.

### B.6 Line Digraphs

For a directed graph $G = (V, E)$, its *Line Digraph* (Harary & Norman, 1960) $G' = (E, P)$ is obtained by taking the original edge set $E$ as node set, and connecting all those new nodes that, as edges in $G$, form a directed path of length two: $P = \{((u, v), (w, x)) | (u, v), (w, x) \in E, v = w\}$.

Both MAGNNETO and our adaptation *M-Slim* operate on the Line Digraph of the network state $S_t$. This means that edge input features like LU and past link weight are provided as node features in $S'_t$ and that *M-Slim* outputs link weight values as node features.

### B.7 PPO

Given the episode length $H = 100$, each training iteration of 16 episodes by default uses 1600 sampled environment transitions to do 10 update epochs with a minibatch size of 400. We multiply the value loss function with a factor of 0.5, clip the gradient norm to 0.5 and use policy and value clip ratios of 0.2 as per Schulman et al. (2018). We use a discount factor of $\gamma = 0.99$ and use $\lambda_{\text{GAE}} = 0.95$ for Generalized Advantage Estimation (Andrychowicz et al., 2020). We model the value function baseline that Proximal Policy Optimization (PPO) uses for variance reduction as separate network that is defined analogous to the respective policy, but uses a mean over all outputs to provide a single value estimate of the global observation. While MAGNNETO is implemented and trained in Tensorflow (Abadi et al., 2015), its PPO algorithm uses the same scaling and clipping parameters except for an additional entropy loss mixin of 0.001.

### B.8 Policy Architectures and Implementation

The original implementation of MAGNNETO uses an MPNN design (Gilmer et al., 2017) implemented in Tensorflow that consists of $L = 8$ message passing layers as described in Bernárdez et al. (2023). Each layer uses a 2-layer MLP with a hidden size 128 and an output size of 16 for the message processing function $m(\cdot)$, a concatenation of `min` and `max` for the message aggregation operation $\oplus$, and a final node feature update $h(\cdot)$ that is realized by a 3-layer MLP with hidden dimensions 128 and 64. The base dimensionality of the feature vectors $h_i$ is 16, meaning that every input and output of the MLP is of that size. At the start of inference, MAGNNETO uses zero-value padding to reach 16 dimensions since the input consists of only two features (LU and past link weights). After $L$ message passing steps, a final readout function yields the actions per edge. They specify which link weights should be incremented by one. The readout function is implemented as a 3-layer MLP with latent dimensions 128 and 64 and dropout layers with a rate of 50% after the first two MLP layers. Except for the final readout layer, the weights for all MLP layers are initialized with an orthogonal matrix with a gain of $\sqrt{2}$, and they are followed by a $\tanh(\cdot)$ activation function.

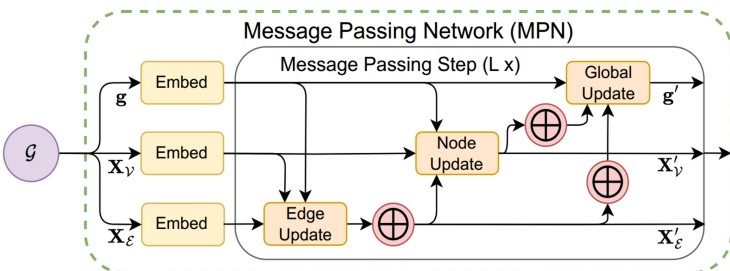

Figure 13: Illustration of the MPN architecture used in our routing approaches *M-Slim* and *FieldLines*.

For *M-Slim* and *FieldLines*, we use a more compact policy architecture leveraging the MPN structure visualized in Figure 13. We use 2 message passing steps, the `mean` and `min` aggregation function in parallel for $\oplus$, and *LeakyReLU* for all activation functions. Moreover, we apply layer normalization (Ba et al., 2016) and residual connections (He et al., 2016) to node and edge features independently after each message passing. For all feature update blocks, we use MLPs with 2 layers and a hidden layer size of 12. In our experiments, the auxiliary distance measure provided to the readout of the *FieldLines*' actor module is the sum of EIGRP link weights for the shortest path from $i$ to $j$. Finally, for the learnable softmax temperature $\tau_\psi$ used during exploration by *FieldLines*'s selector module $\psi$, we use an initial value of 4. For the learnable standard deviation $\sigma_{M\text{-}Slim}$, we use an initial value of 1. We implement our policy modules in PyTorch (Paszke et al., 2019) and use the Adam optimizer with a learning rate of $\alpha = $ 5e-5 (Kingma & Ba, 2014) for *FieldLines*, and 3e-3 for *M-Slim*.

## C   Ablation Studies

In this section we report results for additional experiments that represent ablation studies on our policy design *FieldLines*. Except for Section C.4, all experiments are run on the *nx–XS* topology preset. Default hyperparameter values are mentioned and explained in Section B. We do the ablations on 8 random seeds each, and report results on the better half of them.

### C.1   Learning Settings

Figure 14 shows results for learning hyperparameter ablations. The first two columns per matrix show results for different starting values for the learnable temperature parameter $\tau_\psi$. While a higher starting temperature does not significantly change performance, a lower starting temperature leads to inconsistent improvements and deteriorations across the metrics. The third and fourth columns per matrix show that a notably higher learning rate leads to a collapse in performance, but also that an even lower learning rate is not needed because it does not improve performance. The last two columns show that a lower discount factor does not improve performance, while a higher discount factor incurs minimal performance losses in intense UDP traffic.

|  | UDP, medium traffic | | | | | | UDP, very high traffic | | | | | |
|---|---|---|---|---|---|---|---|---|---|---|---|---|
|  | tau_psi =1 | tau_psi =7 | alpha =1e-5 | alpha =3e-4 | gamma =0.97 | gamma =0.995 | tau_psi =1 | tau_psi =7 | alpha =1e-5 | alpha =3e-4 | gamma =0.97 | gamma =0.995 |
| Received (↑) | -0.01% (14.06MB) | -0.02% (14.06MB) | +0.01% (14.06MB) | -0.09% (14.05MB) | -0.1% (14.04MB) | 0.0% (14.06MB) | -0.08% (51.45MB) | -0.06% (51.46MB) | -0.05% (51.46MB) | -1.79% (50.57MB) | -0.07% (51.45MB) | -0.16% (51.41MB) |
| Avg. Delay (↓) | -0.15% (7.14ms) | +0.04% (7.15ms) | +0.16% (7.16ms) | +1.14% (7.23ms) | -0.08% (7.14ms) | +0.01% (7.15ms) | +0.05% (8.27ms) | +0.12% (8.28ms) | +0.15% (8.28ms) | +13.18% (9.36ms) | +0.06% (8.27ms) | +1.04% (8.36ms) |
| Dropped (↓) | -0.0% (0.14%) | 0.0% (0.14%) | -0.01% (0.14%) | -0.01% (0.13%) | -0.0% (0.14%) | -0.0% (0.14%) | +0.12% (3.26%) | +0.06% (3.19%) | +0.05% (3.18%) | +1.7% (4.83%) | +0.07% (3.2%) | +0.18% (3.31%) |

|  | TCP, medium traffic | | | | | | TCP, very high traffic | | | | | |
|---|---|---|---|---|---|---|---|---|---|---|---|---|
|  | tau_psi =1 | tau_psi =7 | alpha =1e-5 | alpha =3e-4 | gamma =0.97 | gamma =0.995 | tau_psi =1 | tau_psi =7 | alpha =1e-5 | alpha =3e-4 | gamma =0.97 | gamma =0.995 |
| Received (↑) | +0.31% (48.4MB) | -0.17% (48.17MB) | +0.1% (48.3MB) | -7.16% (44.8MB) | +0.15% (48.32MB) | +0.05% (48.28MB) | +0.14% (90.73MB) | +0.03% (90.63MB) | +0.1% (90.69MB) | -19.14% (73.26MB) | +0.05% (90.64MB) | -0.01% (90.59MB) |
| Avg. Delay (↓) | -0.65% (7.18ms) | -0.17% (7.21ms) | -0.12% (7.21ms) | -3.17% (6.99ms) | -0.1% (7.21ms) | -0.14% (7.21ms) | +0.02% (7.48ms) | +0.01% (7.48ms) | -0.05% (7.48ms) | +9.23% (8.17ms) | -0.02% (7.48ms) | -0.01% (7.48ms) |
| Dropped (↓) | -0.04% (2.74%) | -0.0% (2.78%) | 0.0% (2.78%) | +0.37% (3.16%) | -0.01% (2.77%) | +0.01% (2.79%) | -0.02% (9.8%) | +0.01% (9.83%) | -0.0% (9.82%) | +2.95% (12.77%) | 0.0% (9.82%) | 0.0% (9.82%) |

Figure 14: Results for *FieldLines* on the *nx–XS* topology preset when training with different learning hyperparameters as noted in the x-axis labels. The first row of each cell text displays results relative to the base setting, i.e., a *FieldLines* model using $\alpha = $ 5e-5, $\gamma = 0.99$, and $\tau_\psi$ initially set to 4. The second row of the cell text displays absolute numbers. Results show the mean values over 100 evaluation episodes.

## C.2 Architecture Ablations

Figures 15 and16 shows results for architectural ablations on the *FieldLines* policy design. The first two columns of Figure 15 show variations on the latent dimension used by the MLPs within the MPN, which in our main experiments is set to 12. The numbers do not show a clear benefit of either a smaller or a larger latent dimension, but given the worse average delay for UDP traffic of lesser intensity, we keep the latent dimension of 12 even though for the evaluated scenarios a latent dimension of 6 may be enough. The last two columns show results when using either one of the minimum or mean aggregation functions for the permutation-invariant aggregation $\oplus$ of the MPN's node feature update. The overall performances are very similar, such that using e.g. only the mean function for $\oplus$ to reduce complexity is conceivable. Concerning the depth of the used MPN architecture, Figure 16 shows that using $L = 2$ message passing layers provides the best overall results, but the best performing random seeds are very close across all MPN depths. Interestingly, for small graphs and high-volume UDP traffic as well as for larger graphs and medium-volume TCP traffic, average performance across multiple random seeds decreases for MPNs with more layers, because a growing number of seeds fails to converge properly. Also, as the action fluctuation numbers show, adding more message passing layers does not solve *FieldLines*'s problem of increasingly static routing for larger graph topologies.

The *M-Slim* architecture used in this work is considerably smaller than the original MPNN architecture used for MAGNNETO (which is detailed in Section B.8), since this further improves performance while further decreasing inference time. Figure 17 shows results for architectural ablations on *M-Slim*'s policy architecture. The figure's `M-Like` column displays results of an *M-Slim* policy that uses the original MPNN-based policy architecture of MAGNNETO. While being worse than the default architecture for *M-Slim*, it still outperforms MAGNNETO clearly, particularly for TCP traffic. This demonstrates that most of the improvement is achieved by leveraging *PackeRL*'s packet-level dynamics during training. Figure 17 also shows that reducing the number of message passing layers or the latent dimensionality further reduces inference times but compromises routing performance in some traffic settings. On the other hand, increasing the latent dimensionality or layer count increases inference time but does not improve routing performance.

## C.3 Optimizing for Different Objectives

Recent deep RL-powered RO approaches optimize for varying objectives. For example, they have maximized throughput (Fu et al., 2020), or minimized maximum LU (Bernárdez et al., 2023; Chen et al., 2022), packet delay/latency Guo et al. (2022); Sun et al. (2021) or drop counts (Fu et al., 2020). Therefore, to assess the validity of our chosen reward function, Figure 18 presents results for *FieldLines* on the *nx–XS* preset with

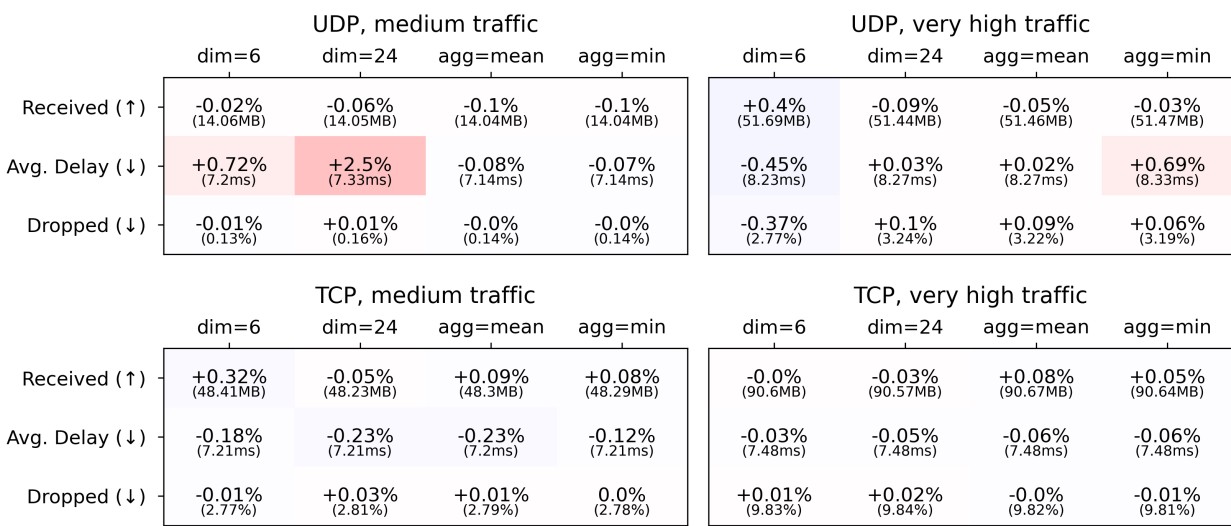

Figure 15: Results for *FieldLines* on the *nx–XS* topology preset when training with policy architecture ablations as noted in the x-axis labels. The first row of each cell text displays results relative to the base setting, i.e., a *FieldLines* model using a concatenation of min and mean aggregation and a latent dimension of 12. The second row of the cell text displays absolute numbers. Results show the mean values over 100 evaluation episodes.

varying optimization objectives. We denote our default reward function describing global goodput in MB by `r`. Let the step-wise penalty function `a` describe the average packet delay in ms, function `d` the ratio of dropped bytes to dropped and received bytes, and function `l` the maximum LU observed in the past step. We can obtain multi-component reward functions by combining these functions. When combined with `r`, we scale the penalty functions `a` by 5 and `d` by 0.25. We found these scaling factors by ensuring that the values lie in the same order of magnitude, meaning that each objective is weighted roughly equally. Figure 18 shows the results per optimization objective, where `rd`, `ra` and `rda` denote the corresponding combinations. Optimizing for the drop ratio (`d`) decreases the drop ratio slightly but may lead to compromises in the other metrics. In turn, optimizing for packet delay (`a`) yields lower delay values, but at the expense of goodput. Interestingly, optimizing solely for LU (`l`) works well in UDP-only traffic, but collapses when dealing with TCP traffic. Finally, the results show large improvements in average delay and drop ratio for the composite optimization objectives in traffic-intense TCP situations, but otherwise show equal or minimally worse performance. All in all, we conclude that optimizing for global goodput alone is a viable objective for our setup. While composite optimization objectives can improve packet delay and drop ratio in some settings, they do not do so consistently and thus do not warrant the additional complexity introduced by multi-component optimization.

## C.4 Training on Different Topologies

Furthermore, we investigate how the choice of training topologies affects *FieldLines*' routing performance. For this, we evaluate three models on the *nx–S* topology preset that have been trained on different topologies. In addition to the default model which is trained on *nx–XS*, we train a model on *nx–S* and another on just the two topologies NSFNet and GEANT2, configured in the same way as in Bernárdez et al. (2023). Figure 19 shows that training on just NSFNet and GEANT2 yields minimally worse results, implying that covering a wide range of different network topologies in the training procedure may be beneficial. On the other hand, training on larger topologies does not improve performance and is therefore not preferred due to longer training times (up to 37 hours to simulate the same amount of training steps).

**UDP, medium traffic, 6-10 nodes**

|  | L=2 | L=1 | L=3 | L=4 |
|---|---|---|---|---|
| Action Fluctuation | 3.65% | -0.12% (1.55, 5.32) | +1.21% (2.5, 8.05) | +0.89% (1.88, 8.07) |
| Received (↑) | 13.91MB | -0.01% (13.9, 13.91) | -0.0% (13.91, 13.91) | 0.0% (13.91, 13.91) |
| Avg. Delay (↓) | 7.11ms | +0.3% (7.11, 7.29) | +0.85% (7.11, 7.37) | +1.18% (7.11, 7.69) |
| Dropped (↓) | 0.13% | 0.0% (0.13, 0.16) | -0.01% (0.09, 0.14) | -0.01% (0.11, 0.13) |

**UDP, medium traffic, 11-25 nodes**

|  | L=2 | L=1 | L=3 | L=4 |
|---|---|---|---|---|
| Action Fluctuation | 1.95% | +0.07% (0.97, 3.05) | +0.43% (1.71, 3.66) | +0.04% (1.19, 2.78) |
| Received (↑) | 31.99MB | -0.02% (31.95, 32.0) | -0.0% (31.99, 32.0) | -0.0% (31.99, 32.0) |
| Avg. Delay (↓) | 10.28ms | +0.56% (10.28, 10.74) | +0.02% (10.28, 10.29) | +0.06% (10.28, 10.33) |
| Dropped (↓) | 0.1% | -0.0% (0.09, 0.1) | 0.0% (0.09, 0.1) | -0.0% (0.09, 0.1) |

**UDP, medium traffic, 26-50 nodes**

|  | L=2 | L=1 | L=3 | L=4 |
|---|---|---|---|---|
| Action Fluctuation | 1.17% | -0.15% (0.47, 1.48) | +0.16% (0.94, 2.11) | -0.13% (0.54, 1.99) |
| Received (↑) | 57.43MB | -0.01% (57.37, 57.43) | -0.0% (57.43, 57.44) | -0.0% (57.43, 57.43) |
| Avg. Delay (↓) | 13.4ms | +0.25% (13.4, 13.66) | +0.01% (13.4, 13.41) | +0.17% (13.4, 13.58) |
| Dropped (↓) | 0.07% | 0.0% (0.07, 0.07) | -0.0% (0.07, 0.07) | 0.0% (0.07, 0.07) |

**UDP, very high traffic, 6-10 nodes**

|  | L=2 | L=1 | L=3 | L=4 |
|---|---|---|---|---|
| Action Fluctuation | 4.55% | -0.33% (2.44, 8.06) | +1.09% (3.33, 8.77) | +2.42% (4.71, 9.41) |
| Received (↑) | 51.61MB | -0.02% (51.51, 51.68) | -2.45% (46.39, 51.64) | -18.98% (30.75, 48.51) |
| Avg. Delay (↓) | 8.16ms | -0.93% (8.05, 8.25) | +7.0% (8.07, 10.09) | +13.02% (7.98, 10.34) |
| Dropped (↓) | 2.76% | 0.0% (2.63, 2.86) | +2.84% (2.67, 14.65) | +20.98% (9.88, 47.9) |

**UDP, very high traffic, 11-25 nodes**

|  | L=2 | L=1 | L=3 | L=4 |
|---|---|---|---|---|
| Action Fluctuation | 3.13% | -0.74% (1.33, 4.21) | -0.59% (1.23, 4.14) | -0.63% (1.76, 4.13) |
| Received (↑) | 111.29MB | -0.14% (110.98, 111.3) | -0.69% (108.47, 111.18) | -1.66% (106.1, 111.0) |
| Avg. Delay (↓) | 12.18ms | -0.03% (12.14, 12.2) | +1.35% (12.17, 12.75) | +2.36% (12.24, 12.99) |
| Dropped (↓) | 9.18% | +0.11% (9.17, 9.41) | +0.64% (9.27, 11.58) | +1.89% (9.43, 15.24) |

**UDP, very high traffic, 51-100 nodes**

|  | L=2 | L=1 | L=3 | L=4 |
|---|---|---|---|---|
| Action Fluctuation | 1.94% | -1.3% (0.35, 1.26) | -1.29% (0.41, 1.09) | -1.25% (0.38, 1.8) |
| Received (↑) | 298.53MB | -0.24% (297.52, 298.64) | -0.32% (297.29, 297.85) | -0.33% (297.49, 297.69) |
| Avg. Delay (↓) | 21.76ms | +0.06% (21.75, 21.8) | -0.08% (21.69, 21.77) | -0.22% (21.68, 21.77) |
| Dropped (↓) | 18.04% | +0.2% (17.97, 18.33) | +0.27% (18.23, 18.39) | +0.28% (18.28, 18.34) |

**TCP, medium traffic, 6-10 nodes**

|  | L=2 | L=1 | L=3 | L=4 |
|---|---|---|---|---|
| Action Fluctuation | 4.23% | -0.96% (1.94, 5.05) | +0.81% (2.49, 7.89) | +1.03% (3.71, 6.71) |
| Received (↑) | 48.6MB | +0.08% (48.6, 48.67) | -2.38% (43.43, 48.69) | -2.76% (44.61, 48.59) |
| Avg. Delay (↓) | 7.02ms | +0.05% (7.0, 7.03) | -1.57% (5.91, 7.11) | +1.43% (6.92, 7.4) |
| Dropped (↓) | 2.7% | -0.0% (2.67, 2.71) | -0.03% (1.83, 3.17) | +0.27% (2.68, 3.97) |

**TCP, medium traffic, 11-25 nodes**

|  | L=2 | L=1 | L=3 | L=4 |
|---|---|---|---|---|
| Action Fluctuation | 2.36% | -0.39% (1.19, 2.54) | +0.63% (1.44, 4.96) | +1.02% (1.75, 8.32) |
| Received (↑) | 86.36MB | +0.07% (86.3, 86.54) | -4.68% (70.31, 86.6) | -3.04% (68.05, 86.37) |
| Avg. Delay (↓) | 9.5ms | +0.11% (9.5, 9.51) | +0.33% (8.52, 10.66) | +0.56% (9.26, 9.84) |
| Dropped (↓) | 3.39% | +0.01% (3.39, 3.44) | +0.3% (2.85, 6.21) | +0.03% (3.38, 3.45) |

**TCP, medium traffic, 51-100 nodes**

|  | L=2 | L=1 | L=3 | L=4 |
|---|---|---|---|---|
| Action Fluctuation | 1.63% | -1.02% (0.36, 1.33) | -0.66% (0.45, 2.69) | -0.09% (0.55, 3.45) |
| Received (↑) | 164.07MB | +0.03% (163.97, 164.31) | -2.31% (141.84, 164.15) | -10.02% (55.59, 164.16) |
| Avg. Delay (↓) | 16.15ms | -0.07% (16.13, 16.14) | -0.39% (14.87, 16.98) | -3.27% (13.32, 16.32) |
| Dropped (↓) | 4.1% | +0.01% (4.1, 4.13) | +0.18% (4.09, 5.23) | +2.65% (3.09, 26.04) |

**TCP, very high traffic, 6-10 nodes**

|  | L=2 | L=1 | L=3 | L=4 |
|---|---|---|---|---|
| Action Fluctuation | 5.7% | +0.69% (3.28, 12.75) | -0.06% (3.27, 7.08) | +0.02% (3.49, 7.77) |
| Received (↑) | 91.28MB | +0.3% (91.44, 92.01) | -0.46% (89.95, 91.52) | -1.46% (88.5, 91.43) |
| Avg. Delay (↓) | 7.29ms | +2.49% (7.34, 7.49) | +2.89% (7.4, 7.58) | -0.01% (6.46, 7.59) |
| Dropped (↓) | 9.39% | +0.4% (9.49, 9.88) | +0.51% (9.66, 10.08) | +0.01% (7.26, 10.16) |

**TCP, very high traffic, 11-25 nodes**

|  | L=2 | L=1 | L=3 | L=4 |
|---|---|---|---|---|
| Action Fluctuation | 3.19% | -0.64% (1.97, 3.54) | -0.47% (1.74, 3.7) | -0.54% (1.47, 4.91) |
| Received (↑) | 154.63MB | +0.23% (154.8, 155.14) | +0.18% (154.62, 155.22) | -0.54% (147.52, 155.1) |
| Avg. Delay (↓) | 9.83ms | +1.29% (9.95, 9.97) | +1.26% (9.93, 9.97) | +1.81% (9.95, 10.35) |
| Dropped (↓) | 11.25% | +0.23% (11.45, 11.53) | +0.25% (11.45, 11.53) | +0.34% (11.48, 12.17) |

**TCP, very high traffic, 51-100 nodes**

|  | L=2 | L=1 | L=3 | L=4 |
|---|---|---|---|---|
| Action Fluctuation | 0.93% | +0.22% (0.52, 3.89) | -0.16% (0.44, 1.23) | -0.06% (0.42, 1.69) |
| Received (↑) | 327.61MB | +0.53% (328.58, 331.76) | +0.35% (328.31, 329.1) | +0.08% (325.22, 328.99) |
| Avg. Delay (↓) | 16.36ms | +0.83% (16.48, 16.56) | +0.77% (16.48, 16.49) | +0.65% (16.23, 16.58) |
| Dropped (↓) | 12.33% | +0.13% (12.36, 12.52) | +0.15% (12.44, 12.5) | +0.14% (12.3, 12.51) |

Figure 16: Results for *FieldLines* with different MPN depths, trained on the *nx–XS* topology preset and evaluated on the *nx–XS*, *nx–S* and *nx–L* presets. Results show the mean values over 100 evaluation episodes. Overall, using $L = 2$ steps yields the best results, but the best performing random seeds are very close across all MPN depths. Also, adding more message passing layers does not solve *FieldLines*'s problem of increasingly static routing for larger graph topologies.

**UDP, medium traffic**

| | M-Slim | dim=6 | dim=24 | L=1 | L=4 | M-Like | MAGNNETO |
|---|---|---|---|---|---|---|---|
| Received (↑) | 13.9MB | -0.0% (13.89, 13.91) | -0.01% (13.89, 13.91) | +0.02% (13.89, 13.91) | +0.01% (13.89, 13.91) | +0.01% (13.89, 13.91) | -0.05% (13.86, 13.91) |
| Avg. Delay (↓) | 7.15ms | -0.04% (7.12, 7.17) | -0.0% (7.12, 7.17) | -0.08% (7.12, 7.17) | -0.1% (7.12, 7.16) | -0.06% (7.12, 7.16) | +1.62% (7.13, 7.79) |
| Dropped (↓) | 0.2% | +0.01% (0.16, 0.28) | +0.01% (0.17, 0.28) | -0.01% (0.15, 0.28) | -0.0% (0.16, 0.27) | -0.01% (0.16, 0.24) | 0.0% (0.18, 0.26) |
| Inference Time (↓) | 11.32ms | -2.34% (10.27, 11.38) | +3.67% (11.19, 12.15) | -21.33% (8.73, 9.21) | +39.39% (15.33, 16.17) | +7.95% (11.73, 13.45) | +1909.57% (226.16, 228.99) |

**UDP, very high traffic**

| | M-Slim | dim=6 | dim=24 | L=1 | L=4 | M-Like | MAGNNETO |
|---|---|---|---|---|---|---|---|
| Received (↑) | 52.02MB | -1.73% (50.02, 52.09) | -0.21% (51.48, 52.11) | -0.0% (51.6, 52.12) | -0.01% (51.72, 52.13) | -0.37% (51.3, 52.12) | -2.39% (48.97, 51.8) |
| Avg. Delay (↓) | 7.93ms | +2.49% (7.9, 8.39) | +0.58% (7.91, 8.17) | +0.21% (7.91, 8.14) | -0.01% (7.91, 7.97) | +0.65% (7.91, 8.16) | +5.96% (8.02, 9.16) |
| Dropped (↓) | 2.14% | +1.57% (2.01, 5.64) | +0.19% (1.99, 3.03) | -0.01% (1.96, 2.83) | +0.01% (1.96, 2.65) | +0.3% (1.96, 3.26) | +2.07% (2.48, 7.25) |
| Inference Time (↓) | 11.41ms | -0.11% (11.15, 11.59) | +4.16% (11.23, 12.25) | -19.7% (8.84, 9.44) | +42.09% (15.64, 16.77) | +7.12% (11.79, 12.51) | +1865.78% (220.15, 227.63) |

**TCP, medium traffic**

| | M-Slim | dim=6 | dim=24 | L=1 | L=4 | M-Like | MAGNNETO |
|---|---|---|---|---|---|---|---|
| Received (↑) | 49.59MB | -2.54% (46.34, 50.47) | -0.36% (46.96, 50.28) | -1.41% (47.03, 50.38) | -0.53% (47.66, 50.31) | -0.72% (48.23, 50.3) | -8.14% (41.58, 49.22) |
| Avg. Delay (↓) | 6.94ms | +0.1% (6.94, 6.96) | +0.02% (6.94, 6.95) | +0.08% (6.94, 6.96) | -0.01% (6.94, 6.95) | -0.06% (6.91, 6.95) | +1.72% (6.93, 7.41) |
| Dropped (↓) | 2.59% | +0.21% (2.53, 3.06) | +0.03% (2.53, 2.97) | +0.13% (2.53, 2.99) | +0.07% (2.53, 2.88) | +0.06% (2.48, 2.81) | +0.56% (2.65, 3.71) |
| Inference Time (↓) | 11.6ms | -2.97% (10.88, 11.6) | +5.1% (11.27, 12.71) | -20.71% (8.9, 9.77) | +42.43% (15.71, 17.04) | +8.28% (12.13, 13.41) | +1908.94% (226.42, 239.03) |

**TCP, very high traffic**

| | M-Slim | dim=6 | dim=24 | L=1 | L=4 | M-Like | MAGNNETO |
|---|---|---|---|---|---|---|---|
| Received (↑) | 92.81MB | -0.94% (88.9, 93.68) | -0.04% (91.15, 93.61) | +0.56% (92.64, 93.76) | -0.24% (90.99, 93.55) | -0.83% (91.16, 93.4) | -9.21% (76.84, 92.4) |
| Avg. Delay (↓) | 7.43ms | -0.25% (7.37, 7.44) | -0.04% (7.39, 7.44) | +0.09% (7.43, 7.44) | -0.05% (7.39, 7.44) | -0.19% (7.37, 7.44) | +1.29% (7.35, 7.82) |
| Dropped (↓) | 9.58% | +0.11% (9.46, 10.1) | +0.03% (9.46, 9.83) | -0.06% (9.45, 9.58) | +0.05% (9.52, 9.89) | +0.14% (9.47, 9.88) | +1.01% (9.71, 11.42) |
| Inference Time (↓) | 11.85ms | -2.04% (10.71, 12.01) | +7.78% (11.6, 13.39) | -17.13% (9.2, 10.35) | +45.85% (15.65, 18.04) | +12.29% (12.64, 13.79) | +1788.96% (219.98, 228.04) |

Figure 17: Results for *M-Slim* on the *nx–XS* topology preset with policy architecture ablations as noted in the x-axis labels. Results show the mean values over 100 evaluation episodes. Overall, the MPN design with default parameters that we use for *M-Slim* slightly improves performance and inference time over using the original MPNN policy architecture of MAGNNETO (column `M-Like`) as explained in Section B.8. Reducing the number of MPN layers or the latent dimensionality further reduces inference times but compromises routing performance in some traffic settings. In any case, the *M-Slim* designs clearly outperform MAGNNETO especially for TCP traffic, showing the importance of packet-level dynamics in training.

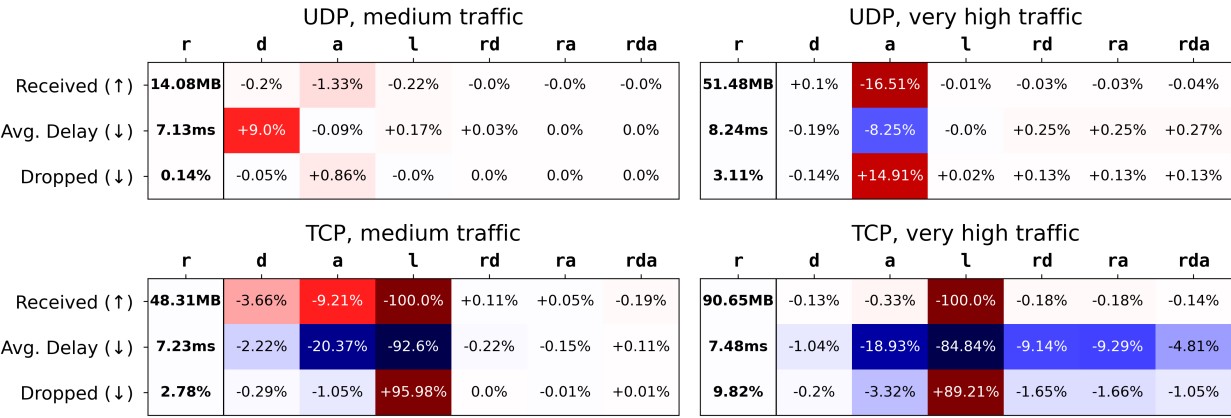

Figure 18: Results for *FieldLines* optimized for different objectives. Results show the mean values over 100 evaluation episodes. Values and colors are relative to the default optimization objective `r` (optimizing for goodput). Letters `d`, `a` and `l` denote optimization for drop ratio, average delay or maximum LU respectively, and concatenated letters denote composite objectives. Optimizing for different objectives influences routing behavior, but no alternative for `r` improves performance consistently.

## C.5 Feature Importance

Finally, Figure 20 shows results for *FieldLines* when restricting or adjusting the policy's access to some features. Overall, while the individual interactions between available features and policy performance are complex, the results generally show that all monitored global and edge features are relevant to some extent, and that removing global or edge features generally decreases performance in some of the evaluated settings. Adding node-level sent/received/retransmitted features (first two columns of the plot) does not improve performance, and in conjunction with a larger latent dimension leads to notable deterioration. Removing edge features, in general, decreases performance in most settings. Interestingly, some edge feature removals can come with improvements in the average delay, and discarding edge features altogether is the next best approach. Concerning global features, all features are relevant to some extent, but the combined removal of

| | UDP, medium traffic | | UDP, very high traffic | | TCP, medium traffic | | TCP, very high traffic | |
| --- | --- | --- | --- | --- | --- | --- | --- | --- |
| | NSFNet +GEANT2 | nx-S | NSFNet +GEANT2 | nx-S | NSFNet +GEANT2 | nx-S | NSFNet +GEANT2 | nx-S |
| Received (↑) | -0.11% (30.33MB) | 0.0% (30.36MB) | -0.26% (109.57MB) | -0.05% (109.8MB) | -0.0% (83.31MB) | +0.06% (83.37MB) | -0.27% (159.8MB) | -0.21% (159.9MB) |
| Avg. Delay (↓) | -0.1% (10.1ms) | +0.07% (10.11ms) | +0.64% (12.66ms) | +1.09% (12.72ms) | +0.16% (9.53ms) | +0.14% (9.53ms) | +0.11% (9.84ms) | +0.01% (9.83ms) |
| Dropped (↓) | -0.01% (0.05%) | -0.0% (0.05%) | +0.17% (9.15%) | +0.05% (9.03%) | -0.0% (3.46%) | 0.0% (3.46%) | +0.05% (11.46%) | +0.01% (11.43%) |

Figure 19: Results for *FieldLines* on the *nx–S* topology preset when training on different topologies as noted in the x-axis labels. The first row of each cell text displays results relative to the base setting, i.e., a *FieldLines* model that is trained on *nx–XS*. The second row of the cell text displays absolute numbers. Results show the mean values over 100 evaluation episodes.

sent/received/dropped/retransmitted bytes and average delay and jitter, or the removal of global load features (link utilization and average datarate utilization) lead to the largest performance penalties. Interestingly, again, removing all global features mitigates a portion of the performance penalties observed when removing parts of the global features. Finally, removing both edge-level and global information about network load (LU and datarate utilization) results in severe performance penalties for UDP traffic, which is also the case when restricting both edge and global features to sent/received/dropped/retransmitted information.

**UDP, medium traffic**

| | all dim=12 | all dim=24 | edge: no SRD | edge: no config | edge: no load | edge: no SRD no config | edge: none | global: no SRDR | global: no DJ | global: no load | global: no SRDR no DJ | global: none | no load | SRD: node only | SRD: edge→node | load only | SRD only | no SRD |
| --- | --- | --- | --- | --- | --- | --- | --- | --- | --- | --- | --- | --- | --- | --- | --- | --- | --- | --- |
| Received (↑) | -0.0% (13.91, 13.91) | -0.01% (13.91, 13.91) | -0.09% (13.81, 13.91) | -0.0% (13.91, 13.91) | -0.01% (13.9, 13.91) | 0.0% (13.91, 13.92) | -0.0% (13.91, 13.92) | -0.0% (13.91, 13.91) | -0.01% (13.91, 13.91) | 0.0% (13.91, 13.92) | -9.02% (4.34, 13.92) | 0.0% (13.91, 13.92) | -0.1% (13.81, 13.91) | 0.0% (13.91, 13.92) | 0.0% (13.91, 13.91) | 0.0% (13.91, 13.92) | -0.07% (13.84, 13.91) | -3.4% (10.13, 13.91) |
| Avg. Delay (↓) | -0.0% (7.11, 7.12) | +2.17% (7.11, 8.0) | +0.55% (7.11, 7.32) | +1.06% (7.11, 7.54) | +0.45% (7.11, 7.27) | +0.64% (7.11, 7.37) | +0.13% (7.11, 7.16) | +0.37% (7.11, 7.32) | +1.53% (7.11, 7.72) | +0.33% (7.11, 7.2) | +8.78% (7.1, 9.85) | +0.47% (7.11, 7.26) | +6.2% (7.11, 10.58) | +0.24% (7.11, 7.25) | +0.18% (7.11, 7.19) | +0.45% (7.11, 7.26) | +0.23% (7.11, 7.18) | +1.03% (7.11, 7.47) |
| Dropped (↓) | 0.0% (0.13, 0.13) | 0.0% (0.12, 0.17) | +0.05% (0.09, 0.58) | -0.01% (0.07, 0.13) | 0.0% (0.1, 0.18) | -0.01% (0.09, 0.13) | -0.0% (0.11, 0.13) | -0.0% (0.09, 0.13) | -0.01% (0.08, 0.13) | -0.01% (0.11, 0.13) | +8.21% (0.08, 63.84) | -0.01% (0.09, 0.13) | +0.08% (0.11, 0.79) | -0.0% (0.1, 0.13) | -0.0% (0.11, 0.13) | -0.01% (0.09, 0.13) | +0.05% (0.12, 0.53) | +2.66% (0.08, 21.49) |

**UDP, very high traffic**

| | all dim=12 | all dim=24 | edge: no SRD | edge: no config | edge: no load | edge: no SRD no config | edge: none | global: no SRDR | global: no DJ | global: no load | global: no SRDR no DJ | global: none | no load | SRD: node only | SRD: edge→node | load only | SRD only | no SRD |
| --- | --- | --- | --- | --- | --- | --- | --- | --- | --- | --- | --- | --- | --- | --- | --- | --- | --- | --- |
| Received (↑) | 0.0% (51.59, 51.63) | -4.87% (44.12, 51.52) | -0.36% (50.47, 51.67) | -1.06% (49.12, 51.59) | -0.51% (50.2, 51.6) | -0.11% (51.35, 51.61) | -0.32% (51.13, 51.58) | -0.1% (51.32, 51.66) | -0.02% (51.59, 51.61) | -8.65% (32.23, 51.59) | -2.77% (45.8, 51.64) | -0.89% (50.04, 51.59) | -5.83% (44.03, 51.62) | -0.16% (51.36, 51.61) | +0.02% (51.58, 51.65) | -1.7% (45.98, 51.62) | -7.87% (40.08, 51.68) | -1.82% (48.03, 51.68) |
| Avg. Delay (↓) | -1.21% (8.06, 8.07) | +15.91% (8.34, 10.93) | +2.15% (8.07, 9.14) | +6.76% (8.07, 10.66) | +3.14% (8.07, 9.57) | +0.41% (8.06, 8.73) | +0.54% (8.01, 9.01) | +0.4% (8.06, 8.57) | -0.47% (8.07, 8.26) | -0.38% (7.68, 9.2) | +11.31% (8.07, 12.88) | +3.12% (7.92, 10.03) | +13.75% (8.07, 11.25) | +0.19% (8.06, 8.5) | +0.19% (8.06, 8.48) | +6.03% (8.06, 11.36) | +18.89% (8.07, 12.18) | +6.03% (8.05, 10.32) |
| Dropped (↓) | -0.01% (2.71, 2.79) | +5.52% (2.82, 18.62) | +0.3% (2.61, 4.82) | +1.03% (2.77, 8.45) | +0.37% (2.76, 4.77) | +0.07% (2.75, 3.1) | +0.27% (2.8, 3.5) | +0.04% (2.66, 3.13) | -0.0% (2.77, 2.79) | +8.77% (2.77, 43.1) | +2.53% (2.68, 12.77) | +0.82% (2.75, 5.13) | +6.27% (2.73, 19.37) | +0.1% (2.75, 3.06) | -0.04% (2.62, 2.81) | +1.6% (2.73, 12.97) | +9.11% (2.63, 28.3) | +1.69% (2.62, 9.85) |

**TCP, medium traffic**

| | all dim=12 | all dim=24 | edge: no SRD | edge: no config | edge: no load | edge: no SRD no config | edge: none | global: no SRDR | global: no DJ | global: no load | global: no SRDR no DJ | global: none | no load | SRD: node only | SRD: edge→node | load only | SRD only | no SRD |
| --- | --- | --- | --- | --- | --- | --- | --- | --- | --- | --- | --- | --- | --- | --- | --- | --- | --- | --- |
| Received (↑) | +0.02% (48.58, 48.66) | -12.66% (26.1, 48.68) | -2.02% (45.28, 48.6) | -0.71% (47.68, 48.61) | -0.32% (48.05, 48.65) | +0.14% (48.61, 48.79) | -0.08% (48.38, 48.65) | -0.69% (46.69, 48.63) | +0.03% (48.56, 48.69) | -0.23% (48.29, 48.61) | -12.68% (17.83, 48.72) | +0.03% (48.47, 48.66) | -0.37% (48.11, 48.73) | -0.55% (47.69, 48.65) | +0.11% (48.6, 48.76) | -0.31% (47.14, 48.71) | -0.85% (46.35, 48.7) | -0.73% (46.46, 48.68) |
| Avg. Delay (↓) | +0.07% (7.02, 7.03) | +4.59% (7.02, 8.1) | -4.01% (5.84, 7.03) | -0.59% (6.86, 7.03) | -0.57% (6.86, 7.03) | -0.01% (6.99, 7.02) | +0.26% (7.02, 7.08) | -0.99% (6.63, 7.03) | +0.1% (7.01, 7.07) | +0.26% (7.02, 7.08) | +4.01% (6.89, 8.35) | -0.18% (6.88, 7.03) | +0.2% (6.92, 7.16) | -1.02% (6.69, 7.05) | -0.06% (6.92, 7.05) | +0.81% (7.02, 7.45) | -0.79% (6.42, 7.16) | +0.47% (6.88, 7.43) |
| Dropped (↓) | +0.01% (2.69, 2.71) | +4.52% (2.65, 20.61) | -0.27% (1.75, 2.72) | -0.05% (2.47, 2.77) | -0.06% (2.46, 2.73) | -0.02% (2.58, 2.71) | +0.01% (2.68, 2.76) | -0.07% (2.36, 2.77) | -0.0% (2.67, 2.71) | +0.01% (2.68, 2.73) | +2.65% (2.59, 20.6) | -0.01% (2.58, 2.71) | -0.01% (2.57, 2.79) | -0.06% (2.33, 2.76) | -0.01% (2.56, 2.72) | +0.01% (2.67, 2.83) | -0.06% (2.12, 2.8) | -0.01% (2.49, 2.9) |

**TCP, very high traffic**

| | all dim=12 | all dim=24 | edge: no SRD | edge: no config | edge: no load | edge: no SRD no config | edge: none | global: no SRDR | global: no DJ | global: no load | global: no SRDR no DJ | global: none | no load | SRD: node only | SRD: edge→node | load only | SRD only | no SRD |
| --- | --- | --- | --- | --- | --- | --- | --- | --- | --- | --- | --- | --- | --- | --- | --- | --- | --- | --- |
| Received (↑) | +0.17% (91.38, 91.63) | -0.5% (88.83, 91.48) | 0.0% (91.05, 91.54) | -0.03% (90.86, 91.48) | +0.08% (91.02, 91.62) | +0.19% (91.38, 91.55) | +0.21% (91.03, 92.23) | +0.02% (90.94, 91.51) | +0.19% (91.3, 91.55) | +0.07% (90.95, 91.51) | -0.43% (89.9, 91.38) | +0.16% (91.32, 91.48) | -0.04% (90.79, 91.43) | -0.06% (90.71, 91.56) | +0.25% (91.09, 92.41) | +0.12% (91.25, 91.48) | +0.18% (91.33, 91.54) | -0.22% (90.24, 91.46) |
| Avg. Delay (↓) | +2.78% (7.49, 7.5) | +3.17% (7.49, 7.6) | +2.54% (7.38, 7.5) | -0.05% (6.73, 7.49) | +1.19% (6.73, 7.5) | +2.65% (7.42, 7.5) | +0.04% (6.17, 7.5) | -0.15% (6.07, 7.5) | +2.78% (7.49, 7.5) | +2.1% (7.33, 7.5) | -1.66% (6.73, 7.49) | +2.37% (7.36, 7.49) | -1.25% (6.06, 7.5) | -1.26% (6.11, 7.5) | +2.5% (7.34, 7.5) | +2.62% (7.4, 7.49) | +0.32% (6.06, 7.5) | -0.83% (6.14, 7.55) |
| Dropped (↓) | +0.47% (9.83, 9.88) | +0.56% (9.86, 10.18) | +0.46% (9.57, 9.95) | -0.04% (7.89, 9.88) | +0.18% (7.82, 9.91) | +0.46% (9.7, 9.88) | -0.05% (6.59, 9.88) | -0.03% (6.53, 9.88) | +0.48% (9.84, 9.94) | +0.37% (9.47, 9.88) | -0.38% (7.6, 9.89) | +0.42% (9.56, 9.93) | -0.24% (6.47, 9.86) | -0.27% (6.58, 9.88) | +0.4% (9.53, 9.87) | +0.45% (9.63, 9.9) | +0.06% (6.52, 9.89) | -0.19% (6.55, 9.98) |

Figure 20: Results for *FieldLines* trained with different available features. Results show the mean values over 100 evaluation episodes. Values and colors are relative to the default feature setting explained in Section B.4. In the column names, SRD abbreviates sent/received/dropped bytes, SRDR abbreviates sent/received/dropped/retransmitted bytes, and DJ abbreviates average packet delay and jitter. The interactions between the policy's access to features and its routing performance are very complex, as the combination of some missing features may result in severe performance issues while others may not have a strong effect. Adding node features and thus using the full set of monitored features does not improve performance, and even leads to large penalties when increasing the latent dimensionality of the policy's MPN.

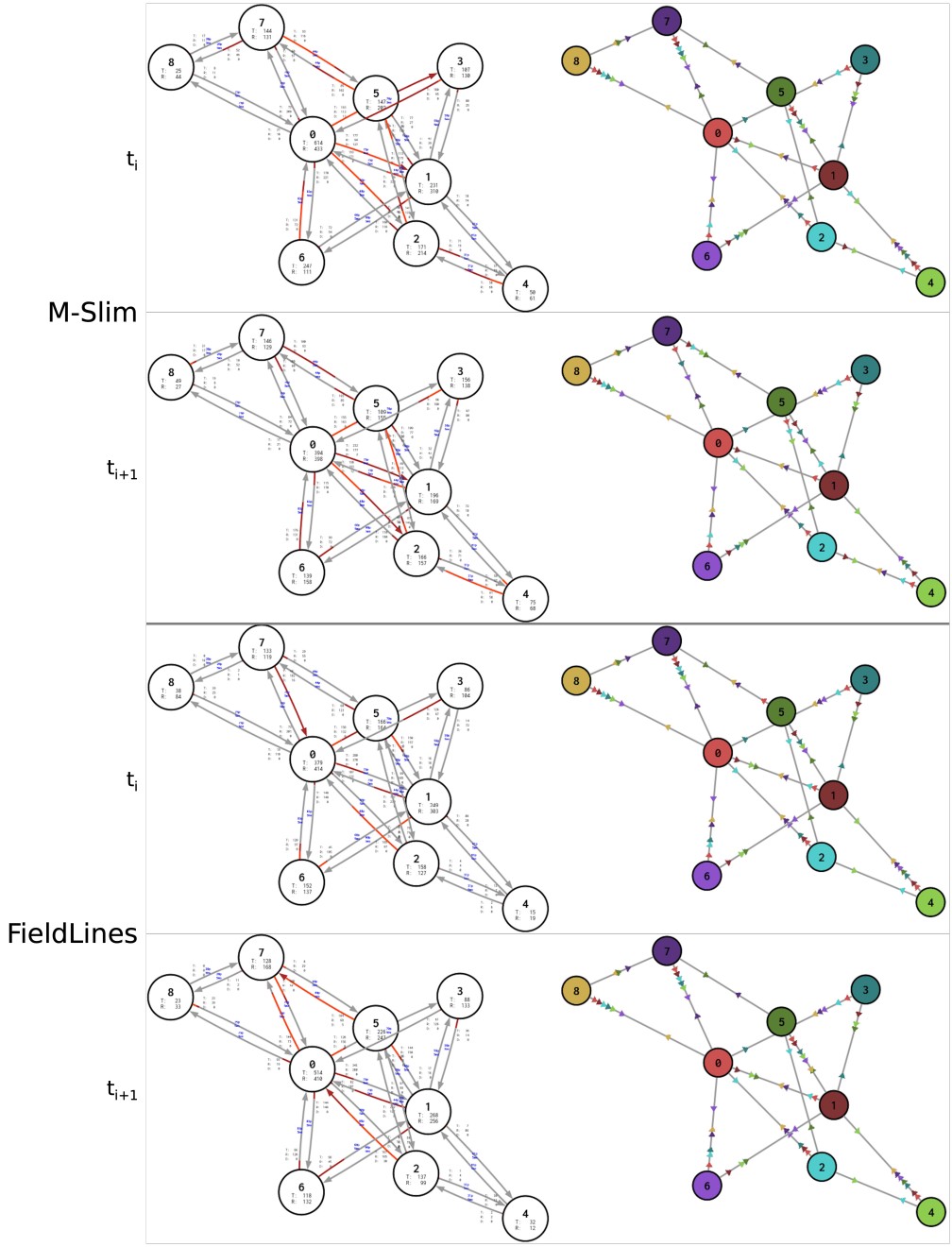

Figure 21: Illustration of network monitoring graphs (left column) and corresponding routing actions (right column) in two consecutive timesteps, taken from the evaluation on the *nx–XS* topology preset of Section 7.1. The dark red color on the monitoring graph's edge illustrations denote the maximum packet buffer fill of the incident network device in the past timestep, the light red color denotes the packet buffer fill at the end of the past timestep (e.g. 50% red = 50% filled). In the action visualization, routing nodes hold distinct colors, and the small colored arrows placed on the edges show where packets destined for the correspondingly colored destination node are sent next. The two upper rows of the figure show network states seen and actions taken by M-Slim, the two lower rows show network states seen and actions taken by *FieldLines*). M-Slim adjusts a few routing selections as relevant edges of the monitoring graph become less congested (e.g. edges $7 \rightarrow 5, 3 \rightarrow 0$ and $0 \rightarrow 2$) from the first to the second timestep. On the other hand, *FieldLines* is much more conservative and only changes the next-hop selection for nodes 0 and 8 at routing node 5, even though e.g. the edges between nodes 0 and 7 have considerably changed in state.

