# OpenReview forum: "Learning Sub-Second Routing Optimization in Computer Networks requires Packet-Level Dynamics"
_TMLR — Accepted by TMLR_

### Review · Reviewer_HPV1 · 2024-08-15

**Summary Of Contributions:**

The paper introduces

1. PackeRL, an augmentation of the existent network routing simulation frameworks
2. two RL-based routing algorithms

   1. M-Slim
   2. FieldLines

3. and a demonstration by simulation that the performance of these algorithms is superior to the existing alternative learned routing algorithms

**Audience:**

Yes

**Broader Impact Concerns:**

There are many concerns raised by this work, but they seem to be already addressed by the Broader Impact Statement.

**Claims And Evidence:**

Yes

**Requested Changes:**

IMO work would be stronger if changes were made to address the weaknesses I raised above, but I don't think these are "dealbreakers"; the paper seems to already represent a publication-worthy body of work. I nonetheless recommend clarifying

**Strengths And Weaknesses:**

I have qualms about my ability to stringently evaluate all the claims made in this paper, as this is far from my area of expertise.
Nonetheless, with the caveat that I am taking many claims at face value, my evaluation is as follows

## Strengths

### PackeRL

Introduces a novel training environment (PackeRL) which seems to expand the range of capabilities of algorithms that learn to solve problems of network traffic routing. This is a non-trivial engineering task, as far as I can see

## M-Slim/FieldLines

Uses this framework to devise two novel packet routing algorithms which have improved performance with respect to various baselines (uniformly-at-random routing, OSPF, ...)

These two things seem to be independently publishable together form a strong basis for the paper

## Weaknesses

### Packet/flow framing

There is an odd structural choice that starts in the title "... Learning Sub-Second Routing Optimization in Computer Networks _requires Packet-Level Dynamics_"

Out of the publicly available environments, only  [  MAGNNETO ] supports arbitrary network
topologies and traffic patterns. But it uses a fluid-based network model, i.e., a model that treats traffic as
flows in a flow network. Such models abstract away from packet-level interactions as encountered in the real
world, e.g. in TCP traffic, where sending and forwarding dynamics are very different from flow distributions
in flow networks.

So in the field of _algorithms which support arbitrary toplogies_ MAGENNTO uses a continuous approximation of a discrete time system of events to perform its modeling, and the title makes it sound like this is the reason that it doesn't work as well as the authors' own M-Slim. IS that what is actually shown in the paper though? According to the authors MAGENNTO is slowed down by its nested loop (e.g. section 7.3 "MAGNNETO’s optimization loop quickly leads to inference times of several seconds" ) which does not seem to be related to the continuous modellng approximation

### Wall-clock modeling

It would be good to have the net computational burden of the routing algorithms *at run time* more transparent. The authors note "FieldLines still requires more time to re-optimize routing than the duration $\tau_\text{sim}$ we simulate per step" which is... good to know, but also suggests that we should be graphing inference time for all algorithms *against $\tau_\text{sim}$*; I understand that this inference-time performance is potentially complicated, but it seems to be a metric of great importance to practical application, so it should be easier to extract this information.

## Evaluation against alternatives

The authors restrict their comparison to only MAGENNTO, on the basis that it is the only approach that generalises to novel topologies. It does nonetheless seem that (Stampa et al., 2017; Pham et al., 2019; Sun et al.) might provide a useful baseline case for fixed-topology networks.

---

> ### Author Response · Authors · 2024-08-20
> **Reply to Reviewer HPV1**
>
> Thank you very much for your review. The concerns you have raised help us to improve the presentation quality and clarity of the paper, especially with respect to its scope. We will integrate the following clarifications into an updated version of the paper text.
>
> ### Packet/Flow Framing
>
> It is true that the nested loop of MAGNNETO and the resulting high inference times are not directly related to our claim that packet-level dynamics in training are important. Rather, we support our claim by showing the difference in performance between policies trained in fluid-based and packet-level environments, and the high inference times of MAGNNETO pose an obstacle that we need to overcome by doing away with the nested loop (which yields _M-Slim_). The high inference times of MAGNNETO are problematic because training in PackeRL involves many more temporally fine-grained interaction steps than MAGNNETO’s environment, which coarsely aggregates traffic into Traffic Matrices. Therefore, the direct inference of _M-Slim_ is not only a feature aside our main claim, but a necessary prerequisite to support the claim, because it makes training in _PackeRL_ computationally feasible to begin with.
>
> ### Wall-clock modeling
>
> This is a good point which invites us to clarify the role of inference time in our problem setting. In the current framework, as commonly done in RL research, environment execution (network simulation) is halted during action selection. This is equivalent to the hypothetical case where the network simulation runs continuously, and action inference and installation times are zero. The fact that FieldLines’s inference time is higher than what happens to be the value of $\tau_{\text{sim}}$ is, indeed, nice to know but not essential. Instead, inference time becomes important in the even more realistic setting where network operation would continue during action inference. Learning sub-second routing optimization with RL, to the best of our knowledge, has not yet been studied in such an environment. We will expand on this in future work, since implementing the option for continuous simulation requires some non-trivial adjustments to _PackeRL_.
>
> ### Evaluation Scope and Baselines
>
> We fully agree that additional strong baselines could further back our argument. While a policy hand-crafted or fine-tuned for a specific network topology may possibly provide even better performance, we chose to not evaluate topology-specific approaches. This is because, in the wild, network topology changes can occur involuntarily, and given the high temporal frequency of our routing interactions, the ability to deal with different topologies without the need for re-training is essential. MAGNNETO has shown that learning useful general-purpose representations for routing optimization is possible, not least because they are also somewhat capable of routing in PackeRL. In our evaluation, we expand on this finding by showing the importance of packet-level dynamics for learning better representations.

---

### Review · Reviewer_BM42 · 2024-08-26

**Summary Of Contributions:**

This paper details the construction of a new network routing
simulation environment, PackeRL, which simulates network flows and
allows training RL route optimization networks with packet-level
simulation. This is in contrast to fluid-based models, which lack the
finer granularity of packet-level simulation. The authors also train
two new RL routing networks using this environment: M-Slim (a
modification of an existing method MAGNNETO) and FieldLines, both
GNNs. The authors observe better routing performance (in particular
goodput) when trained using the packet data, and the architectures
allow for faster evaluation allowing, routing decisions to be made
more frequently.

**Audience:**

Yes

**Broader Impact Concerns:**

No specific concerns

**Claims And Evidence:**

Yes

**Requested Changes:**

First, a few cases where I think additional discussion would enhance
clarity and help make the paper a bit more self-contained:

1. I would appreciate more details on MAGNNETO and the changes made in
   constructing M-Slim--architecturally as well as in how the networks
   are evaluated.

2. Overall, more details on the architectures I think would help as
   well. I did see the discussion of latent dimensions, etc but I
   think being a bit more explicit on the architectures would help
   readers (even with the future release of source code).

3. And one question just for clarity: it appears that the training
   topologies in each nx-(XS,S,...) etc are generated once and then fixed.
   Is the same true at evaluation and true of the traffic events at
   evaluation time? (Just to understand which aspects of the evaluation
   episodes were the same for each set of networks.) It might help to
   briefly mention this alongside the training and evaluation details
   in section 6.2.

Then a few questions I have: some discussion of these might help
readers, but I think is certainly less critical:

1. At a somewhat quick glance, it looks like MAGNNETO carried out a
   sort of distributed evaluation while the process here might be more
   centralized (i.e. it appears inference is carried out while the
   simulation is paused and is therefore instantaneous w.r.t. the
   packet events). Is this correct? Is this another difference from
   MAGNNETO, or does the fact that simulation used in that work lacks
   packet-level behavior reduce their ability to account for this
   effect?

2. If that is correct do you have some sense of whether accounting for
   the costs of inference (possibly lagging the application of the
   networks' routing decisions) would change the behavior
   significantly? I suppose this may intersect with a possible future
   extension to account for age-of-information while performing
   inference.

**Strengths And Weaknesses:**

**Strengths**

- The packet-level simulation environment appears very useful, with more realistic behavior than fluid-based models
- Good detail is provided on the configuration of simulation tasks, generation of topologies, and simulated events
- Experiments on proposed RL methods include varying topologies and
  traffic compositions (UDP vs. TCP mixes, intensity, etc.)

**Weaknesses**

- The paper could use more detail on the architectures of the two
  methods, M-Slim and FieldLines, and a clearer discussion of changes
  from MAGNNETO (in the case of M-Slim)
- Some ablation tests conducted are only carried out over a very limited
  range of parameters, making trends somewhat less clear (although the
  costs of the RL training do seem make to these already expensive to
  conduct)

---

> ### Author Response · Authors · 2024-08-30
> **Reply to Reviewer BM42**
>
> We thank you sincerely for your review. Your remarks are very valid, and in responding to them we hope to further improve the paper’s presentation. Addressing your comments by topic:
>
> ### Architectural/Evaluation details, changes from MAGNNETO
>
> We agree that the architectures of the evaluated approaches and their differences, together with their evaluation procedure, should be explained in more detail. We update the text of Sections 5, 6 and Appendix B to include more information.
>
> ### Ablation Study Scope
>
> We acknowledge that the individual ablation studies highlight trends with a rather limited set of parameter combinations. As you have pointed out, training multiple random seeds for every possible parameter combination and evaluating them on a variety of individual traffic setups is resource-intensive. We are working on extending our ablations by investigating on feature importance and the role of GNN depth.
>
> ### Network Scenarios
>
> Network scenarios are freshly generated at the start of each training/evaluation episode. They contain the network topology as well as traffic demand and optional link failure events for the entire simulation span of the episode. We can configure synnet to generate different kinds of network scenarios for training and evaluation, e.g. _nx—XS_ topologies with low UDP traffic for training, and _nx—S_ topologies with high TCP traffic + link failures for evaluation. Supplying different random seeds varies the generated scenarios, while using the same seed ensures reproducibility across multiple training/evaluation runs. We clarify this aspect in the text of section 6.
>
> ### Distributed vs. Centralized Evaluation, Inference Costs
>
> This is an interesting point. The authors of Bernardez et al. (2023), claim that MAGNNETO is a distributed multi-agent system, but apart from modeling suggestions they do not provide experimental results for distributed routing in their paper. Instead, the public implementation [1] uses single-agent PPO for training, trains and evaluates in a fully centralized manner, and pauses its environment during action selection. This is equally true for _M-Slim_, _FieldLines_ and the current interaction with _PackeRL_. Indeed, this raises the question of the importance of inference time in larger networks, as well as interaction settings where the network does not stop during action selection. We update Section 8.1 to include this clarification.
>
>
> [1] https://github.com/BNN-UPC/MAGNNETO-TE

---

> > ### Comment · Reviewer_BM42 · 2024-09-27
> >
> > Thank you very much for your thorough responses to my questions. I
> > believe that your revisions have strengthened the paper. I also
> > especially appreciate your insights into the distributed vs
> > centralized evaluation question and your closer knowledge of the
> > magnneto implementation.

---

### Review · Reviewer_QwUa · 2024-08-27

**Summary Of Contributions:**

The paper investigates reinforcement learning (RL) - based routing algorithms for sub-second routing optimization in computer networks. The authors argue that standard routing protocols such as OSPF or EIGRP may not be able to exploit the monitored information (including events such as link failures) as effectively as RL-based approaches, On the other hand, RL-based approaches can be computationally more expensive; furthermore, most prior work on such methods relies on fluid network models that do not accurately capture the complex nature of packet-level dynamics (e.g., via the UDP or TCP protocols). To fill this gap in the literature, the authors introduce PackeRL, a packet-level RL environment for routing in generic network topologies. To enable sub-second routing optimization, they additionally propose two algorithms, namely, M-Slim and FieldLines. M-Slim is a shortest path algorithm that performs very strongly with high traffic (especially TCP), but is hard to scale to larger networks. Its lack of scalability is addressed by the second algorithm, FieldLines, which relies on a next-hop (as opposed to shortest path) policy design. To assess the effectiveness of the proposed algorithms, the authors conduct an extensive empirical study, on PackeRL which corroborates their various claims.

**Audience:**

Yes

**Broader Impact Concerns:**

No concern.

**Claims And Evidence:**

Yes

**Requested Changes:**

I am open to changing my score, but I would want to understand what the authors think about all aforementioned weaknesses, If there are points I have misunderstood, I would appreciate additional clarifications.

Furthermore, something I found a bit confusing in the various tables was that the authors provide only the absolute changes. Wouldn't it have been better to show the percentage changes? I was curious as to why the authors made this particular presentation choice. I actually had to work out the percentage changes myself on many occasions, which was inefficient.

**Strengths And Weaknesses:**

Strengths
1. The proposed packet-level simulation environment with RL-powered routing optimization is a nice contribution that addressed the various gaps in the existing literature. In the absence of real testbeds, the next best thing is accurate simulators. Flow-level simulators do not take into account the various intricacies, subtleties, and complexities of real routing protocols, as opposed to packet-level simulators. In that sense, PackeRL is an important addition that can be used for a more fair and precise benchmark evaluation.
2. I like the fact that the authors propose two different algorithms, each with different strengths and weaknesses, and possibly different applicability. This is in fact quite aligned with standard routing protocols, where we observe a significant variety (e.g., OSPF, BGP, EIGRP, etc.). Having a variety of choices is a strength, depending on the application scenario.
3. The overall framework is quite flexible, as it can accept different reward definitions, state configurations, traffic types,, network topologies, etc.
4. The experimental study is extensive and quite through. The results indicate that the proposed methods have potential as competitors not only to other ML-based frameworks but also other traditional routing protocols. M-Slim, in particular, seems to perform very strongly with high traffic. Also, compared to other ML methods, inference time for M-Slim and, in particular, FieldLines is significantly lower, which makes the framework more practical.

Weaknesses
I have concerns on several fronts. In detail:
1. The results are somehow mixed. Let's for example focus on the table in Figure 3. In the TCP/UDP mixed traffic regime, M-Slim reduces average delay by 0.24/7.2=3.33%. But it increases the percentage of dropped packets by 0.15/1.25=12%, while it reduces the number of received MB by 0.13/35.38=0.37%. Similarly, in the same setting, FieldLines reduces average delay by 0.16/7.2=2.22%, but increases the percentage of dropped packets by 0.14/1.25=11.4%. In the TCP high regime, the results are arguably better for the two methods, and this is again seen in Figure 4, where for high and very high traffic the two introduced algorithms fare better than EIGRP. But for low or medium traffic, the results are bad or mixed in the best case. To me this creates various challenges:
- First, note that the real competitor of the proposed algorithm is not random algorithms or other ML-based algorithms, but standard routing protocols. This is the comparison that matters the most, if the goal of this work is to argue that the proposed framework is a viable competitor for practical routing in real computer networks.
- Second, the fact that performance depends so critically on the nature of the traffic could be an issue. In particular, I find it rather distressing that both algorithms give mixed or even negative results for low or medium traffic. If the goal is for these algorithms to be deployed to real networks, then we would ideally want them to perform strongly under all settings. Traffic naturally fluctuates, so performing strongly under a narrow regime of traffics may be a weakness.
- Actually, the aforementioned point may be related to the optimization objective: if, for instance, we want to minimize delay or maximize throughout, RL may find it beneficial to increased the number of dropped packets under some circumstances. On the other hand, standard routing protocols are based on strong intuitions, heuristics or combinatorial algorithms, which have been found to be very effective in practice.
- Overall, the mixed results provide at the very best mixed evidence about the potential of RL-based approaches to act as substitutes for the traditional routing protocols.

2. I am confused by the POMDP formulation. In deep learning, one could argue that we feed the raw data to the algorithm, and RL will learn over time to extract the features that are important for optimizing the chosen reward function. On the other hand, the authors adopt a POMDP formulation, where only a subset of the features are chosen. But doesn't that complicate the problem formulation unnecessarily? What do the authors gain from this? At the end of the day, MDP (rather than POMDP) approaches are used. So, why introduce a complex formulation, and not simply let the system learn the correct representation on its own?

3. I am not clear whether this is the best venue for such a work. The packet-level simulator and the innovations therein are great but not of interest to the ML community. The ML/RL part of this work is what is potentially of interest. But, as I previously wrote, I am not sure this work provides strong evidence that the RL-based framework is good under all traffic settings. If the goal of the authors was simply to argue that their algorithms have some potential but without totally solving the routing problem, then a networking venue would have been more appropriate in my view. Given the actual machine learning contribution is not pronounced, I think an ML audience would at least want to see more compelling empirical evidence. I was quite surprised that the authors decided to leave some key RL aspects (e.g., reward function design and/or scalarization, multi-objective RL, etc.) as future work, even though these are precisely the aspects that an ML audience would be more interested in.

4. I was not so clear about the role of $\tau_{sim}$, when it comes to standard optimization strategies. In principle, couldn't protocols like EIGRP or OSPF deal with much lower values for $\tau_{sim}$? I mean, can't we run them several times in the the period of $\tau_{sim}$, given their comparatively lower computational cost? For example, in the high traffic regime, do the authors currently run, say, EIGRP only once in the simulation period $\tau_{sim}$,? What if they ran it several times, which should be feasible given this protocol is more lightweight? Wouldn't then EIGRP adapt better to the high-volume traffic? In general, I was not clear how the standard protocols interact with the simulation period $\tau_{sim}$, and how many times each method is run within that period. For the same reason, I was not clear whether it is fair to give the same $\tau_{sim}$ to, say, EIGRP, or whether it can afford an even lower value due to its lower computational cost.

---

> ### Author Response · Authors · 2024-08-30
> **Reply to Reviewer QwUa**
>
> Thank you very much for your review. It helps us to clarify the scope and importance of our work, and improve on its presentation quality. We address your concerns by topic:
>
> ### Results; Venue and Importance to ML Community
>
> We agree that submitting to an ML venue requires providing valuable insights for the ML community. In our view, this work contains the insights needed to enable and motivate further research on sub-second routing in computer networks as an ML problem. With our experiments we have shown the versatility of PackeRL and the importance of leveraging packet-level dynamics to learn better representations for routing optimization. This makes it a good contender for a benchmark environment, which given the fragmented state of the RL-for-RO community may be of great use. Concerning the proposed RL algorithms, we acknowledge that they do not improve routing in all evaluated scenarios. Simply using “shortest” paths obtained from heuristics seems to be an optimal or close-to-optimal strategy in low-traffic settings. Yet, our policies show that we can do better in the more challenging and interesting high-volume traffic setting, and that the learned representations work in any topology. Practically, one could already implement a rule-based system switching between heuristics and M-Slim/FieldLines depending on traffic conditions to get the best of both worlds. Finally, we believe that our framework, training procedure and policy designs will inspire advances in ML-powered routing for other types of networks like power grids or road networks, exposing and leveraging structural similarities.
>
> ### POMDP Formulation
>
> This is a valid point. We have set out to include situations with partial observability in our formalism, but as the paper stands, we agree that the problem is better modeled as an MDP. Simply put, we know the entire network state because we monitor it, and it is the policy’s task to learn representations from the important features. We update our formalism and related descriptions in Section 3.
>
> ### $\tau_{\text{sim}}$ and OSPF/EIGRP
>
> In our setup, we pause network simulation during action selection for all considered approaches as commonly done in RL research. This enables us to simulate the network with the latest routing actions for a pre-defined amount of time $\tau_{\text{sim}}$ per step, regardless of the inference time required by the respective routing algorithm. As in the discussion with Reviewer HPV1, inference time becomes important in the "real-time" setting where network operation would continue during action inference, which may be an interesting direction for future work.
>
> Concerning EIGRP and OSPF, by default, they adapt to changing topologies but not to changing traffic conditions and network utilization. This implies that they only need to be run at the start of the episode and once the network topology changes, but also that running them more often during simulation does not improve the routing. This explanation is missing in section 6, and we update the text accordingly.
>
> ### Reporting Absolute vs. Percentage Changes in Figures
>
> We fully agree that showing percentage changes for the amount of received data and the average packet delay makes it easier to assess performance. The changes in the reported metrics are currently displayed in “absolute” changes because the “Dropped” metric reports a percentage already. For the low drop percentage values encountered in some of our evaluation settings, reporting the percentage change of the “Dropped” percentage would greatly overstate the importance, especially when compared to the other metrics. Anyhow, we will adjust our plots accordingly to show percentage changes for the other metrics.

---

> > ### Comment · Reviewer_QwUa · 2024-08-31
> >
> > I thank the authors for their detailed responses. Overall, my concerns were addressed. A general comment after reading all reviews and comments is that it is not so clear from this work whether the proposed algorithms will achieve the same good results in real testbeds and networks, where we cannot simply pause the network to recompute an optimal policy. The authors are transparent about this in Section 8.1. But the concern remains: to what extent can we hope to deploy such algorithms in real networks, where speed and real-time considerations are very critical? Is it possible for example that given all the time, hardware, and protocol constraints, the more traditional protocols would be preferred in practice?
> >
> > Furthermore, despite the good gains in high-traffic settings, results in low-traffic settings are often suboptimal. Yes, I agree that we can have a simple rule: use traditional protocols for low-traffic settings; else, use RL. But this is not a very convincing argument. AlphaGo, for instance, does not distinguish between high-ELO and low-ELO components; it can just handle any of them. This dichotomy between the low- and high-traffic regimes is perhaps something the authors would want to reconsider in the future.
> >
> > All that being said, I agree this work can be viewed as a potentially very important application of RL to networking, with encouraging results. For this reason, I will update my score.

---

### Author Response · Authors · 2024-08-30
**General Response to Reviewers**

We thank the reviewers for their useful and insightful comments. They have already helped to increase the presentation quality of our work substantially and make it more self-contained. We have uploaded a first update of the paper that contains the improvements and clarifications from the discussions up to now. In particular, we have made the following adjustments:

- [Reviewer QwUa, Section 3.1] Turning the POMDP formalism into an MDP formalism (includes minor renames throughout the paper),
- [Reviewer HPV1, Sections 5.1, 6 and 6.1] Adding more detail on evaluation framing and goal, and the necessity of improving MAGNNETO’s run time (the discussion of evaluated approaches has moved to a new Section 6.1),
- [Reviewer BM42, Sections 5, 5.1, 5.2, B.6, B.7 and B.8] Adding more detail on the architectures of the considered RL policies and their common points and differences,
- [Reviewer BM42, Sections 6 and 6.2] Clarifying the episodic setup for training and evaluation, where each episode comes with its own network scenario,
- [Reviewer BM42, Section 6.1] Adding more detail on how the learned policies are evaluated,
- [Reviewer QwUa, Section 6.1] Clarifying strengths (flexible wrt. topology) and weaknesses (oblivious wrt. traffic) of heuristic baselines,
- [Reviewer HPV1, Section 8.1] Expanding on the importance of inference time and our current limitation on non-real-time routing,
- [Reviewer BM42, Section 8.1] Adding that the learned policies were trained and evaluated in a centralized manner,
- [Reviewer QwUa, all figures] Reporting relative changes for goodput and packet delay for better readability (we have unified the color scales across metrics, since they are all reported as percentages now).

The updated paper text is marked with a reviewer-specific color (Reviewer HPV1: orange, Reviewer BM42: magenta, Reviewer QwUa: teal) for illustration purposes. We will remove the text colors again in the final version of the paper.

We are eager to resolve any points that remain unclear.

---

### Author Response · Authors · 2024-09-09
**General Response to Reviewers**

Dear Reviewers,

we would like to inform you that we have updated our submission with the following adjustments:

- Cleaning up the list of references, fixing minor spelling and capitalization mistakes and publication venues for some references,
- Adding further ablation studies on the policy architectures used for _M-Slim_ and _FieldLines_, including investigations on the effect of increasing or decreasing GNN depth,
- Adding ablation studies for _FieldLines_ on the importance of the monitored features.

We will gladly respond to any remaining questions or concerns.

---

### Author Response · Authors · 2024-10-14
**Camera-Ready Version Available**

We would like to thank you again for your valuable input during the review process, which has helped to further increase the quality of our submission.

The camera-ready version of our submission is now available, including a link to the project webpage. Our code has now entered our own internal publication process and will be made available shortly.

Kind regards,
Paper Authors

---

### Decision · Action_Editor_A7C1 · 2024-10-08

**Recommendation:** Accept as is

**Comment:**

The reviewers find the proposed packet-level simulation environment is valuable to tackle more realistic network topologies, and also believe the proposed M-Slim/FieldLines algorithms are quite useful for RL-based routing optimization. After rebuttal, most of the raised concerns have been properly addressed, and two reviewers voted to accept this paper while one reviewer leaned toward accepting.

I read the paper in detail and totally agree with all reviewers that this work has made a solid contribution (a packet-level simulation environment and two powerful algorithms) to routing optimization in computer networks, the claims made in this paper are well supported, and some TMLR audience will find this work useful. Therefore, I am happy to accept this work as is.

In addition, I also agree with the reviewers that testing the proposed algorithms on different real networks could be very helpful to fully demonstrate their pros and cons in a practical scenario. The proposed future extenstions such as multipath and/or multi-objective routing optimization is also crucial for a good RL-for-RO benchmark environment.

**Audience:**

All reviewers believe some individuals in TMLR's audience could be interested in the findings of this paper.

**Claims And Evidence:**

This work investigates the RL-based approach for sub-second routing optimization in computer networks. It aims to address the key issue that no existing RL-powered routing optimization (RO) approach can provide both a realistic simulation backend and a comprehensive toolset for addresssing a wide ranges of realistic network cenarios. The main contributions are two-fold: 1) PackeRL: the first packet-level RL environment for routing optimization with arbitrary realistic network topologies, 2) M-Slim and FieldLines: two novel and efficient algorithms for RL-based sub-second routing optimization. The efficiency of the proposed PackeRL environment and M-Slim/FieldLines are well demonstrated by a set of comprehensive experiments.

All reviewers believe the claims made in this paper are supported by accurate, convincing and clear evidence.